# LIPSCHITZ SINGULARITIES IN DIFFUSION MODELS

**Zhantao Yang**[1,4]⋆    **Ruili Feng**[2,4]⋄    **Han Zhang**[1,4]⋆    **Yujun Shen**[3]    **Kai Zhu**[2,4]

**Lianghua Huang**[4]    **Yifei Zhang**[1,4]⋆    **Yu Liu**[4]    **Deli Zhao**[4]    **Jingren Zhou**[4]    **Fan Cheng**[1]†

[1]Shanghai Jiao Tong University
[2]University of Science and Technology of China    [3]Ant Group    [4]Alibaba Group

```
{ztyang196, ruilifengustc, hzhang9617, shenyujun0302}@gmail.com
{zkzy}@mail.ustc.edu.cn  {xuangen.hlh}@alibaba-inc.com
qidouxiong619@sjtu.edu.cn  {ly103369}@alibaba-inc.com
zhaodeli@gmail.com  jingren.zhou@alibaba-inc.com  chengfan@sjtu.edu.cn
```

## ABSTRACT

Diffusion models, which employ stochastic differential equations to sample images through integrals, have emerged as a dominant class of generative models. However, the rationality of the diffusion process itself receives limited attention, leaving the question of whether the problem is well-posed and well-conditioned. In this paper, we explore a perplexing tendency of diffusion models: they often display the infinite Lipschitz property of the network with respect to time variable near the zero point. We provide theoretical proofs to illustrate the presence of infinite Lipschitz constants and empirical results to confirm it. The Lipschitz singularities pose a threat to the stability and accuracy during both the training and inference processes of diffusion models. Therefore, the mitigation of Lipschitz singularities holds great potential for enhancing the performance of diffusion models. To address this challenge, we propose a novel approach, dubbed E-TSDM, which alleviates the Lipschitz singularities of the diffusion model near the zero point of timesteps. Remarkably, our technique yields a substantial improvement in performance. Moreover, as a byproduct of our method, we achieve a dramatic reduction in the Fréchet Inception Distance of acceleration methods relying on network Lipschitz, including DDIM and DPM-Solver, by over 33%. Extensive experiments on diverse datasets validate our theory and method. Our work may advance the understanding of the general diffusion process, and also provide insights for the design of diffusion models.

## 1    INTRODUCTION

The rapid development of diffusion models has been witnessed in image synthesis (Ho et al., 2020; Song et al., 2020; Ramesh et al., 2022; Saharia et al., 2022; Rombach et al., 2022; Zhang & Agrawala, 2023; Hoogeboom et al., 2023) in the past few years. Concretely, diffusion models construct a multi-step process to destroy a signal by gradually adding noises to it. That way, reversing the diffusion process (*i.e.*, denoising) at each step naturally admits a sampling capability. In essence, the sampling process involves solving a reverse-time stochastic differential equation (SDE) through integrals (Song et al., 2021b).

Although diffusion models have achieved great success in image synthesis, the rationality of the diffusion process itself has received limited attention, leaving the open question of whether the problem is well-posed and well-conditioned. In this paper, we surprisingly observe that the noise-prediction (Ho et al., 2020) and v-prediction (Salimans & Ho, 2022) diffusion models often exhibit a perplexing tendency to possess infinite Lipschitz of network with respect to time variable near the zero point. We provide theoretical proofs to illustrate the presence of infinite Lipschitz constants

---

† Corresponding author, ⋆ Work performed at Alibaba Academy, ⋄ Project leader

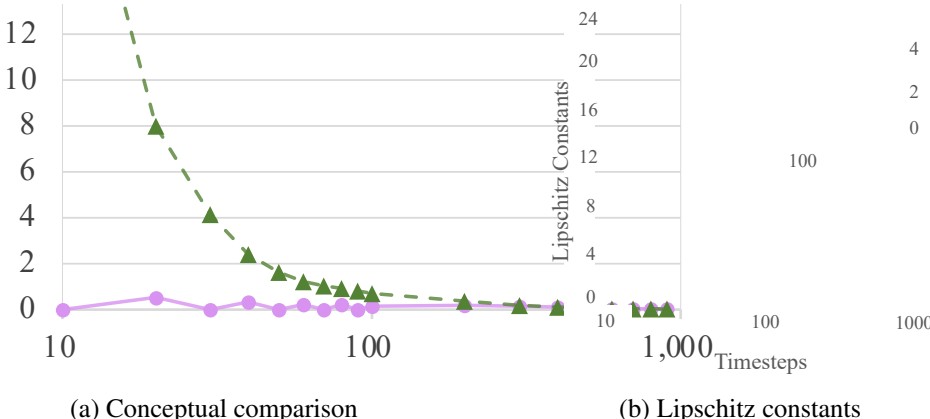

(a) Conceptual comparison        (b) Lipschitz constants

Figure 1: (a) **Conceptual comparison** between DDPM (Ho et al., 2020) (I) and our proposed early timestep-shared diffusion model (E-TSDM) (II). DDPM trains the network $\epsilon_\theta(\cdot, t)$ with varying timestep conditions $t$ at each denoising step, whereas E-TSDM uniformly divides the near-zero timestep interval $t \in [0, \tilde{t})$ with high Lipschitz constants into $n$ sub-intervals and **shares the condition $t$ within each sub-interval**. Here, $\tilde{t}$ denotes the length of the interval for sharing conditions. When $t \geq \tilde{t}$, E-TSDM follows the same procedure as DDPM. However, when $t < \tilde{t}$, E-TSDM shares timestep conditions. (b) **Quantitative comparison** of the Lipschitz constants between DDPM and our proposed early timestep-shared diffusion model (E-TSDM). **The Lipschitz constants tend to be extremely large near zero point for DDPM**. However, our sharing approach allows E-TSDM to force the Lipschitz constants in each sub-interval to be zero, thereby **reducing the overall Lipschitz constants** in the timestep interval $t \in [0, \tilde{t})$, where $\tilde{t}$ is set as a default value 100.

and empirical results to confirm it. Given that noise prediction and v-prediction are widely adopted by popular diffusion models (Dhariwal & Nichol, 2021; Rombach et al., 2022; Ramesh et al., 2022; Saharia et al., 2022; Podell et al., 2023), the presence of large Lipschitz constants is a significant problem for the diffusion model community.

Since we uniformly sample timesteps for both training and inference processes, large Lipschitz constants *w.r.t.* time variable pose a significant threat to both training and inference processes of diffusion models. When training, large Lipschitz constants near the zero point affect the training of other parts due to the smooth nature of the network, resulting in instability and inaccuracy. Moreover, since inference requires a smooth network for integration, the large Lipschitz constants probably have a substantial impact on accuracy, particularly for faster samplers. Therefore, the mitigation of Lipschitz singularities holds great potential for enhancing the performance of diffusion models.

Fortunately, there is a simple yet effective alternative solution: by sharing the timestep conditions in the interval with large Lipschitz constants, the Lipschitz constants can be set to zero. Based on this idea, we propose a practical approach, which uniformly divides the target interval near the zero point into $n$ sub-intervals, and uses the same condition values in each sub-interval, as shown in Figure 1 (II). By doing so, this approach can effectively reduce the Lipschitz constants near $t = 0$ to zero. To validate this idea, we conduct extensive experiments, including unconditional generation on various datasets, acceleration of sampling, and super-resolution task. Both qualitative and quantitative results confirm that our approach substantially alleviates the large Lipschitz constants near zero point and improves the synthesis performance compared to the DDPM baseline (Ho et al., 2020). We also compare this simple approach with other potential methods to address the challenge of large Lipschitz constants, and find our method outperforms all of these alternative methods. In conclusion, in this work, we theoretically prove and empirically observe the presence of Lipschitz singularities issue near the zero point, advancing the understanding of the diffusion process. Besides, we propose a simple yet effective approach to address this challenge and achieve impressive improvements.

## 2 RELATED WORK

The significant advancements of diffusion models have been witnessed in recent years in the domain of image generation. (Karras et al., 2022; Lu et al., 2022b; Dockhorn et al., 2021; Bao et al., 2022b; Lu et al., 2022a; Bao et al., 2022a; Zhang et al., 2023). It (Sohl-Dickstein et al., 2015; Ho et al.,

2020; Song et al., 2021b) defines a Markovian forward process $\{\mathbf{x}_t\}_{t\in[0,T]}$ that gradually destroys the data $\mathbf{x}_0$ with Gaussian noise. For any $t \in [0, T]$, the conditional distribution $q_{0t}(\mathbf{x}_t|\mathbf{x}_0)$ satisfies

$$q_{0t}(\mathbf{x}_t|\mathbf{x}_0) = \mathcal{N}\left(\mathbf{x}_t|\alpha_t\mathbf{x}_0, \sigma_t^2\mathbf{I}\right), \tag{1}$$

where $\alpha_t$ and $\sigma_t$ are referred to as the noise schedule, satisfying $\alpha_t^2 + \sigma_t^2 = 1$. Generally, $\alpha_t$ decreases from 1 to 0 as $t$ increases, to ensure that the marginal distribution of $\mathbf{x}_t$ gradually changes from the data distribution $q_0(x_0)$ to Gaussian. Kingma et al. (2021) further prove that the following stochastic differential equation (SDE) has the same transition distribution $q_{0t}(\mathbf{x}_t|\mathbf{x}_0)$ as in Equation (1) for any $t \in [0, T]$:

$$d\mathbf{x}_t = f(t)\mathbf{x}_t dt + g(t) d\mathbf{w}_t, \quad \mathbf{x}_0 \sim q_0(x_0), \tag{2}$$

where $\mathbf{w}_t$ is the standard Wiener process, $f(t) = \frac{d\log\alpha_t}{dt}$ and $g(t) = 2\sigma_t^2\frac{d\log(\sigma_t/\alpha_t)}{dt}$.

Song et al. (2021b) point out that the following reverse-time SDE has the same marginal distribution $q_t(\mathbf{x}_t)$ for any $t \in [0, T]$:

$$d\mathbf{x}_t = [f(t)\mathbf{x}_t - g(t)^2 \nabla_{\mathbf{x}_t}\log q_t(\mathbf{x}_t)]dt + g(t)d\bar{\mathbf{w}}_t, \quad \mathbf{x}_T \sim q_T(\mathbf{x}_T), \tag{3}$$

where $\bar{\mathbf{w}}_t$ is a standard Wiener process in the reverse time. Once the score function $\nabla_{\mathbf{x}_t}\log q_t(\mathbf{x}_t)$ is known, we can simulate Equation (3) for sampling. However, directly learning the score function is problematic, as it involves an explosion of training loss when having a small $\sigma_t$(Song et al., 2021b). In practice, the noise prediction model $\boldsymbol{\epsilon}_\theta(\mathbf{x}_t, t)$ is often adopted to estimate $-\sigma_t\nabla_{\mathbf{x}_t}\log q_t(\mathbf{x}_t)$. The network $\boldsymbol{\epsilon}_\theta(\mathbf{x}_t, t)$ can be trained by minimizing the objective:

$$\mathcal{L}(\theta) := \mathbb{E}_{t\sim\mathcal{U}(0,T),\mathbf{x}_0\sim q_0(\mathbf{x}_0),\epsilon\sim\mathcal{N}(0,\mathbf{I})}\left[\|\boldsymbol{\epsilon}_\theta(\alpha_t\mathbf{x}_0 + \sigma_t\epsilon, t) - \epsilon\|_2^2\right]. \tag{4}$$

In this work, our observation of Lipschitz singularities on noise-prediction and v-prediction diffusion models reveals the inherent price of such an approach.

**Numerical stability near zero point.** Achieving numerical stability is essential for high-quality samples in diffusion models, where the sampling process involves solving a reverse-time SDE. Nevertheless, numerical instability is frequently observed near $t = 0$ in practice (Song et al., 2021a; Vahdat et al., 2021). To address this singularity, one possible approach is to set a small non-zero starting time $\tau > 0$ in both training and inference (Song et al., 2021a; Vahdat et al., 2021). Kim et al. (2022) resolve the trade-off between density estimation and sample generation performance by introducing randomization to the fixed $\tau$. In contrast, we enhance numerical stability by reducing the Lipschitz constants to zero near $t = 0$, which leads to improved sample quality in diffusion models. It is worth noting that the numerical issues observed by aforementioned works are mainly caused by the singularity of transition kernel $q_{0t}(\mathbf{x}_t|\mathbf{x}_0)$. This transition kernel will degrade to a Dirac kernel $\delta(\mathbf{x}_t - \alpha_t\mathbf{x}_0)$ as $\sigma_t \to 0$. However, our observation is *the infinite Lipschitz constants of the noise prediction model* $\boldsymbol{\epsilon}_\theta(\mathbf{x}, t)$ *w.r.t time variable* $t$, and this is caused by the explosion of $\frac{d\sigma_t}{dt}$ as $t \to 0$. To the best of our knowledge, this has not been observed before.

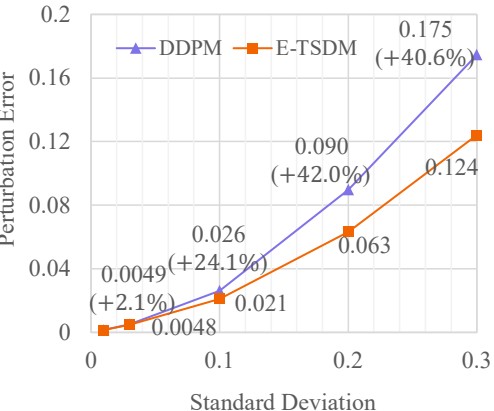

Figure 2: **Quantitative comparison** of the errors caused by a perturbation on the input between E-TSDM and DDPM (Ho et al., 2020). Results show that **E-TSDM is more stable**, as its prediction is less affected, *e.g.*, the perturbation error of DDPM is 42.0% larger than E-TSDM when the perturbation scale is 0.2.

## 3 LIPSCHITZ SINGULARITIES IN DIFFUSION MODELS

**Lipschitz singularities issue.** In this section, we elucidate the vexing propensity of diffusion models to exhibit infinite Lipschitz near the zero point. We achieve this by analyzing the partial derivative $\partial\boldsymbol{\epsilon}_\theta(\mathbf{x}, t)/\partial t$ of the network $\boldsymbol{\epsilon}_\theta(\mathbf{x}, t)$. In essence, the emergence of Lipschitz singularities, characterized by $\limsup_{t\to 0+}\left\|\frac{\partial\boldsymbol{\epsilon}_\theta(\mathbf{x},t)}{\partial t}\right\| \to \infty$, can be attributed to the fact that the prevailing noise schedules conform to the behavior of $d\sigma_t/dt \to \infty$ as the parameter $t$ tends towards zero.

**Theoretical analysis.** Now we theoretically prove that the infinite Lipschitz happens near the zero point in diffusion models, where the distribution of data is an arbitrary complex distribution. We focus particularly on the scenario where the network $\epsilon_\theta(\mathbf{x}, t)$ is trained to predict the noises added to images (v-prediction model (Salimans & Ho, 2022) has a similar singularity problem, and is analyzed in Appendix C.2). The network $\epsilon_\theta(\mathbf{x}, t)$ exhibits a relationship with the score function $\nabla_\mathbf{x} \log q_t(\mathbf{x})$ that $\epsilon_\theta(\mathbf{x}, t) = -\sigma_t \nabla_\mathbf{x} \log q_t(\mathbf{x})$ (Song et al., 2021b), where $\sigma_t$ is the standard deviation of the forward transition distribution $q_{0t}(\mathbf{x}|\mathbf{x}_0) = \mathcal{N}(\mathbf{x}; \alpha_t \mathbf{x}_0, \sigma_t^2 \mathbf{I})$. Specifically, $\alpha_t$ and $\sigma_t$ satisfy $\alpha_t^2 + \sigma_t^2 = 1$.

**Theorem 3.1** *Given a noise schedule, since $\sigma_t = \sqrt{1 - \alpha_t^2}$, we have $\frac{d\sigma_t}{dt} = -\frac{\alpha_t}{\sqrt{1-\alpha_t^2}} \frac{d\alpha_t}{dt}$. As t gets close to 0, the noise schedule requires $\alpha_t \to 1$, leading to $d\sigma_t/dt \to \infty$ as long as $\frac{d\alpha_t}{dt}\big|_{t=0} \neq 0$. The partial derivative of the network can be written as*

$$\frac{\partial \epsilon_\theta(\mathbf{x}, t)}{\partial t} = \frac{\alpha_t}{\sqrt{1 - \alpha_t^2}} \frac{d\alpha_t}{dt} \nabla_\mathbf{x} \log q_t(\mathbf{x}) - \frac{\partial \nabla_\mathbf{x} \log q_t(\mathbf{x})}{\partial t} \sigma_t. \tag{5}$$

*Note that $\alpha_t \to 1$ as $t \to 0$, thus if $\frac{d\alpha_t}{dt}\big|_{t=0} \neq 0$, and $\nabla_\mathbf{x} \log q_t(\mathbf{x})|_{t=0} \neq \mathbf{0}$, then one of the following two must stand*

$$\lim_{t \to 0+} \sup \left\| \frac{\partial \epsilon_\theta(\mathbf{x}, t)}{\partial t} \right\| \to \infty; \quad \lim_{t \to 0+} \sup \left\| \frac{\partial \nabla_\mathbf{x} \log q_t(\mathbf{x})}{\partial t} \sigma_t \right\| \to \infty. \tag{6}$$

Note that $\frac{d\alpha_t}{dt}\big|_{t=0} \neq 0$ stands for a wide range of noise schedules, including linear, cosine, and quadratic schedules (see details in Appendix C.1). Besides, we can safely assume that $q_t(\mathbf{x})$ is a smooth process. Therefore, we may often have $\limsup_{t \to 0+} \left\| \frac{\partial \epsilon_\theta(\mathbf{x}, t)}{\partial t} \right\| \to \infty$, indicating the infinite Lipschitz constants around $t = 0$.

**Simple case illustration.** Take a simple case that the distribution of data $p(\mathbf{x}_0) \sim \mathcal{N}(\mathbf{0}, \mathbf{I})$ for instance, the score function for any $t \in [0, T]$ can be written as

$$\nabla_\mathbf{x} \log q_t(\mathbf{x}) = \nabla_\mathbf{x} \log \left( \frac{1}{\sqrt{2\pi}} \exp\left( -\frac{\|\mathbf{x}\|_2^2}{2} \right) \right) = -\mathbf{x}. \tag{7}$$

Due to the relationship $\epsilon_\theta(\mathbf{x}, t) = -\sigma_t \nabla_\mathbf{x} \log q_t(\mathbf{x})$ and the fact that the deviation $\frac{d\sigma_t}{dt}$ tends toward $\infty$ as $t \to 0$, we have $\left\| \frac{\partial \epsilon_\theta(\mathbf{x}, t)}{\partial t} \right\| \to \infty$.

**Case in reality.** After theoretically proving that diffusion models suffer infinite Lipschitz near the zero point, we show it empirically. We estimate the Lipschitz constants of a network by

$$K(t, t') = \frac{\mathbb{E}_{\mathbf{x}_t}[\|\epsilon_\theta(\mathbf{x}_t, t) - \epsilon_\theta(\mathbf{x}_t, t')\|_2]}{\Delta t}, \tag{8}$$

where $\Delta t = |t - t'|$. For a network $\epsilon_\theta(\mathbf{x}_t, t')$ of DDPM baseline (Ho et al., 2020) trained on FFHQ $256 \times 256$ (Karras et al., 2019) (see training details in Section 5.1 and more results of the Lipschitz constants $K(t, t')$ on other datasets in Appendix D.1), the variation of the Lipschitz constants $K(t, t')$ as the noise level $t$ varies is seen in Figure 1b, showing that the Lipschitz constants $K(t, t')$ get extremely large in the interval with low noise levels. Such large Lipschitz constants support the above theoretical analysis and pose a threat to the stability and accuracy of the diffusion process, which relies on integral operations.

## 4 MITIGATING LIPSCHITZ SINGULARITIES BY SHARING CONDITIONS

**Proposed method.** In this section, we propose the Early Timestep-shared Diffusion Model (E-TSDM), which aims to alleviate the Lipschitz singularities by sharing the timestep conditions in the interval with large Lipschitz constants. To avoid impairing the network's ability, E-TSDM performs a stepwise operation of sharing timestep condition values. Specifically, we consider the interval near the zero point suffering from large Lipschitz constants, denoted as $[0, \tilde{t})$, where $\tilde{t}$ indicates the length of the target interval. E-TSDM uniformly divides this interval into $n$ sub-intervals represented as a sequence $\mathbb{T} = \{t_0, t_1, \cdots, t_n\}$, where $0 = t_0 < t_1 < \cdots < t_n = \tilde{t}$ and $t_1 - t_0 = t_i -$

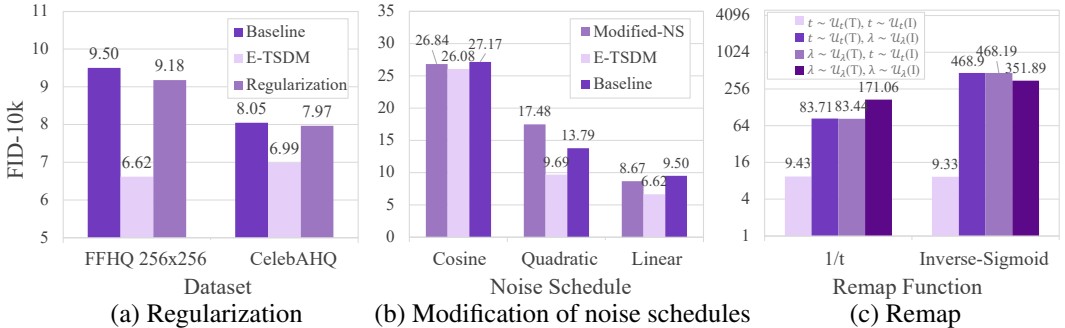

Figure 3: **Quantitative analysis of alternative methods** evaluated with FID-10k ↓. (a) **Regularization**: Experimental results on FFHQ $256 \times 256$ and CelebAHQ $256 \times 256$ show that **regularization techniques can slightly improve the FID of DDPM (Ho et al., 2020) baseline but performs worse than E-TSDM** (b) **Modification of noise schedules (Modified-NS)**: We implement Modified-NS on linear, quadratic, and cosine schedules. Experimental results on FFHQ $256 \times 256$ dataset indicate that the performance of **Modified-NS is unstable while E-TSDM achieves better synthesis performance**. (c) **Remap**: **Quantitative comparison** of **remap** method between uniformly sampling $t$ and uniformly sampling $\lambda$, during training and inference, on FFHQ $256 \times 256$. Specifically, $\mathcal{U}_t$ is $\mathcal{U}[0, 1]$, and $\mathcal{U}_\lambda$ is $\mathcal{U}[0, K]$ for $1/t$ but $\mathcal{U}[-K, K]$ for Inverse-Sigmoid, where $K$ is a large number to avoid infinity. (T) represents the sampling strategy during the training process while (I) represents that during the inference process. Results show that **remap is not helpful**.

$t_{i-1}, \forall i = 1, 2, \cdots, n$. For each sub-interval, E-TSDM employs a single timestep value (the left endpoint of the sub-interval) as the condition, both during training and inference. Utilizing this strategy, E-TSDM effectively enforces zero Lipschitz constants within each sub-interval, with only the timesteps located near the boundaries of the sub-intervals having a Lipschitz constant greater than zero. As a result, the overall Lipschitz constants of the target interval $t \in [0, \tilde{t})$ are significantly reduced. The corresponding training loss can be written as

$$\mathcal{L}\left(\epsilon_\theta\right) := \mathbb{E}_{t \sim \mathcal{U}(0,T), \mathbf{x}_0 \sim q(\mathbf{x}_0), \epsilon \sim \mathcal{N}(0,\mathbf{I})}\left[\left\|\epsilon_\theta\left(\alpha_t \mathbf{x}_0 + \sigma_t \epsilon, f_{\mathbb{T}}\left(t\right)\right) - \epsilon\right\|_2^2\right], \tag{9}$$

where $f_{\mathbb{T}}(t) = \max_{1 \leq i \leq n}\{t_{i-1} \in \mathbb{T} : t_{i-1} \leq t\}$ for $t < \tilde{t}$, while $f_{\mathbb{T}}(t) = t$ for $t \geq \tilde{t}$. The corresponding reverse process can be represented as

$$p_\theta\left(\mathbf{x}_{t-1}|\mathbf{x}_t\right) = \mathcal{N}\left(\mathbf{x}_{t-1}; \frac{\alpha_{t-1}}{\alpha_t}\left(\mathbf{x}_t - \frac{\beta_t}{\sigma_t}\epsilon_\theta\left(\mathbf{x}_t, f_{\mathbb{T}}\left(t\right)\right)\right), \eta_t^2\mathbf{I}\right), \tag{10}$$

where $\beta_t = 1 - \frac{\alpha_t}{\alpha_{t-1}}$, and $\eta_t^2 = \beta_t$. E-TSDM is easy to implement, and the algorithm details are provided in Appendix B.2.

**Analysis of estimation error.** Then we show that the estimation error of E-TSDM can be bounded by an infinitesimal, and thus the impact of E-TSDM on the estimation accuracy is insignificant. The detailed proof is shown in Appendix C.3.

**Theorem 4.1** *Given the chosen $f_{\mathbb{T}}(t)$, when $t \in [0, \tilde{t})$, the difference between the optimal $\epsilon_\theta(\mathbf{x}, f_{\mathbb{T}}(t))$ denoted as $\boldsymbol{\epsilon}^*(\mathbf{x}, f_{\mathbb{T}}(t))$, and $\boldsymbol{\epsilon}(\mathbf{x}, t) = -\sigma_t \nabla_{\mathbf{x}} \log q_t(\mathbf{x})$, can be bounded by*

$$\left\|\boldsymbol{\epsilon}^*\left(\mathbf{x}, f_{\mathbb{T}}\left(t\right)\right) - \boldsymbol{\epsilon}\left(\mathbf{x}, t\right)\right\| \leq \sigma_{\tilde{t}} K\left(\mathbf{x}\right) \Delta t + B\left(\mathbf{x}\right) \Delta\sigma_{\max}, \tag{11}$$

*where*

$$K\left(\mathbf{x}\right) = \sup_{t \neq \tau} \frac{\left\|\nabla_{\mathbf{x}} \log q_t\left(\mathbf{x}\right) - \nabla_{\mathbf{x}} \log q_\tau\left(\mathbf{x}\right)\right\|}{|t - \tau|}, \quad B\left(\mathbf{x}\right) = \sup_t \left\|\nabla_{\mathbf{x}} \log q_t\left(\mathbf{x}\right)\right\|, \tag{12}$$

*and $\Delta\sigma_{\max} = \max_{1 \leq i \leq n} |\sigma_{t_i} - \sigma_{t_{i-1}}|$. Note that $K(\mathbf{x})$ and $B(\mathbf{x})$ are finite and $\lim_{\Delta t \to 0} \Delta\sigma_{\max} = 0$ for any continuous $\sigma_t$ where $\Delta t := \tilde{t}/n$, thus the difference converges to 0 as $\Delta t \to 0$. Furthermore, the rate of convergence is at least $\frac{1}{2}$-order with respect to $\Delta t$.*

The $\frac{1}{2}$-order convergence rate is relatively fast in optimization. Given this bound, we think the introduced errors of E-TSDM are controllable.

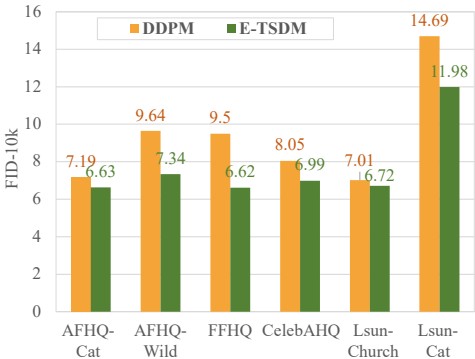

Figure 4: **Quantitative comparison** on various datasets with $256 \times 256$ resolution. All experiments are evaluated with FID-10k $\downarrow$.

Table 1: **Quantitative comparison** on FFHQ (Karras et al., 2019). $*$ denotes our reproduced result with the same network as E-TSDM-large.

| Model | FID-50k $\downarrow$ |
|---|---|
| StyleGAN2+ADA+bCR (Karras et al., 2020) | 3.62 |
| Soft-Truncation (Kim et al., 2021) | 5.54 |
| P2-DM (Choi et al., 2022) | 6.92 |
| LDM (Rombach et al., 2022) | 4.98 |
| DDPM (Ho et al., 2020) | 6.88$*$ |
| E-TSDM (ours) | 5.21 |
| E-TSDM-large (ours) | **4.22** |

**Reduction in Lipschitz constants.** In Figure 1b, we present the curve of $K(t, t')$ of E-TSDM on FFHQ $256 \times 256$ (Karras et al., 2019) (we provide results for continuous-time diffusion models and more results on other datasets in Appendix D.1), showing that the Lipschitz constants $K(t, t')$ are significantly reduced by applying E-TSDM.

**Improvement in stability.** To further verify the stability of E-TSDM, we evaluate the impact of a small perturbation added to the input. Specifically, we add a small noise with a growing scale to the $\mathbf{x}_{\tilde{t}}$, where $\tilde{t}$ is set to a default value of 100, and observe the resulting difference in the predicted value of $\mathbf{x}_0$, for both E-TSDM and baseline. Our results, as shown in Figure 2, illustrate that E-TSDM exhibits better stability than the baseline, as its predictions are less affected by perturbations.

**Comparison with some alternative methods.** Although achieving impressive performance as detailed in Section 5, E-TSDM introduces no modifications to the network architecture or loss function, thereby not incurring any additional computational cost. **1) Regularization**: In contrast, an alternative potential approach is imposing restrictions on the Lipschitz constants via regularization techniques. It necessitates the computation of $\frac{\partial \boldsymbol{\epsilon}_\theta(\mathbf{x}, t)}{\partial t}$, consequently diminishing training efficiency. **2) Modification of noise schedules**: Furthermore, E-TSDM preserves the forward process unaltered. Conversely, another potential method involves the modification of noise schedules. Recall that the issue of Lipschitz singularities only arises when the noise schedule satisfies $\frac{d\alpha_t}{dt}|_{t=0} \neq 0$. Therefore, it becomes feasible to adjust the noise schedule to meet the requirement $\frac{d\alpha_t}{dt}|_{t=0} = 0$, thus mitigating the problem of Lipschitz singularities. The detailed methods for modifying noise schedules are provided in Appendix D.3.2. Although this modification seems feasible, it results in tiny amounts of noise at the beginning stages of the diffusion process, leading to inaccurate predictions. **3) Remap**: In addition, remap is another possible method, which designs a remap function $\lambda = f(t)$ as the conditional input of the network, namely, $\boldsymbol{\epsilon}_\theta(\mathbf{x}, f(t))$. By carefully designing $\lambda = f(t)$, it can significantly stretch the interval with large Lipschitz constants. For example, $f(t) = 1/t$ and $f^{-1}(\lambda) = \text{sigmoid}(\lambda)$ are two simple choices. In this way, Remap can efficiently reduce the Lipschitz constants regarding the conditional inputs of the network, $\frac{\partial \boldsymbol{\epsilon}_\theta(\mathbf{x}, t)}{\partial \lambda}$. However, since we uniformly sample $t$ both in training and inference, what should be focused on is the Lipschitz constants regarding $t$, $\frac{\partial \boldsymbol{\epsilon}_\theta(\mathbf{x}, t)}{\partial t}$, which can not be influenced by remap. We also consider the situation of uniformly sampling $\lambda$, which can significantly hurt the quality of generated images. We show the quantitative evaluation in Figure 3 and put the detailed analysis in Appendix D.3.3. Empirically, E-TSDM surpasses not only the baseline but also all of these alternative methods, where the results are demonstrated in Figure 3. For a more in-depth discussions, please refer to Section D.3.

## 5 EXPERIMENTS

In this section, we present compelling evidence that our E-TSDM outperforms existing approaches on a variety of datasets. To achieve this, we first detail the experimental setup used in our studies in Section 5.1. Subsequently, in Section 5.2, we compare the synthesis performance of E-TSDM against that of the baseline on various datasets. In Section 5.3, we conduct multiple ablation studies and quantitative analysis from two perspectives. Firstly, we demonstrate the generalizability of

Table 2: **Quantitative comparison** between DDPM baseline (Ho et al., 2020) and our proposed E-TSDM on both discrete-time and continuous-time scenarios with different noise schedules, on FFHQ 256 × 256 (Karras et al., 2019) using FID-10k ↓ as the evaluation metric. Experimental results illustrate that **E-TSDM can be generalized to other noise schedules and is still effective in the context of continuous-time diffusion models**.

| Settings | Method | Noise schedule | | | | |
|---|---|---|---|---|---|---|
| | | Linear | Quadratic | Cosine | Cosine-shift | Zero-terminal-SNR |
| Discrete | Baseline | 9.50 | 13.79 | 27.17 | 14.51 | 11.66 |
| | E-TSDM | **6.62** | **9.69** | **26.08** | **11.20** | **8.58** |
| Continuous | Baseline | 9.53 | 14.26 | 25.65 | 12.80 | 10.89 |
| | E-TSDM | **6.95** | **9.66** | **16.80** | **9.94** | **8.96** |

E-TSDM by implementing it on continuous-time diffusion models and varying the noise schedules. Secondly, we investigate the impact of varying the number of conditions $n$ in $t \in [0, \tilde{t})$ and the length of the interval $\tilde{t}$, which are important hyperparameters. Moreover, we demonstrate in Section 5.4 that our method can be effectively combined with popular fast sampling techniques. Finally, we show that E-TSDM can be applied to conditional generation tasks, such as super-resolution, in Section 5.5.

## 5.1 EXPERIMENTAL SETUP

**Implementation details.** All of our experiments utilize the settings of DDPM (Ho et al., 2020) (see more details in Appendix B.1). Besides, we utilize a more developed structure of unet (Dhariwal & Nichol, 2021) than that of DDPM (Ho et al., 2020) for all experiments containing reproduced baseline. Given that the model size is kept constant, the speed and memory requirements for training and inference for both the baseline and E-TSDM are the same. Except for the ablation studies in Section 5.3, all other experiments fix $\tilde{t} = 100$ for E-TSDM and use five conditions ($n = 5$) in the interval $t \in [0, \tilde{t})$, which we have found to be a relatively good choice in practice. Furthermore, all experiments are trained on NVIDIA A100 GPUs. **Datasets.** We implement E-TSDM on several widely evaluated datasets containing FFHQ 256 × 256 (Karras et al., 2019), CelebAHQ 256 × 256 (Karras et al., 2017), AFHQ-Cat 256 × 256, AFHQ-Wild 256 × 256 (Choi et al., 2020), Lsun-Church 256 × 256 and Lsun-Cat 256 × 256 (Yu et al., 2015). **Evaluation metrics.** To assess the sampling quality of E-TSDM, we utilize the widely-adopted Frechet inception distance (FID) metric (Heusel et al., 2017). Additionally, we use the peak signal-to-noise ratio (PSNR) to evaluate the performance of the super-resolution task.

## 5.2 SYNTHESIS PERFORMANCE

We have demonstrated that E-TSDM can effectively mitigate the large Lipschitz constants near $t = 0$ in Figure 1 b, as detailed in Section 4. In this section, we conduct a comprehensive comparison between E-TSDM and DDPM baseline (Ho et al., 2020) on various datasets to show that E-TSDM can improve the synthesis performance. The quantitative comparison is presented in Figure 4, which clearly illustrates that E-TSDM outperforms the baseline on all evaluated datasets. Furthermore, as depicted in Appendix D.5, the samples generated by E-TSDM on various datasets demonstrate its ability to generate high-fidelity images. Remarkably, to the best of our knowledge, as shown in Table 1, we set a new state-of-the-art benchmark for diffusion models on FFHQ 256 × 256 (Karras et al., 2019) using a large version of our approach (see details in Appendix B.1).

## 5.3 QUANTITATIVE ANALYSIS

In this section, we demonstrate the generalizability of E-TSDM by implementing it on continuous-time diffusion models and varying the noise schedules. In addition, to gain a deeper understanding of the properties of E-TSDM, we investigate the critical hyperparameters of E-TSDM by varying the length of the interval $\tilde{t}$ to share the timestep conditions, and the number of sub-intervals $n$.

### 5.3.1 QUANTITATIVE ANALYSIS ON THE GENERALIZABILITY OF E-TSDM

To ensure the generalizability of E-TSDM beyond specific settings of DDPM (Ho et al., 2020), we conduct a thorough investigation of E-TSDM on other popular noise schedules, as well as implement

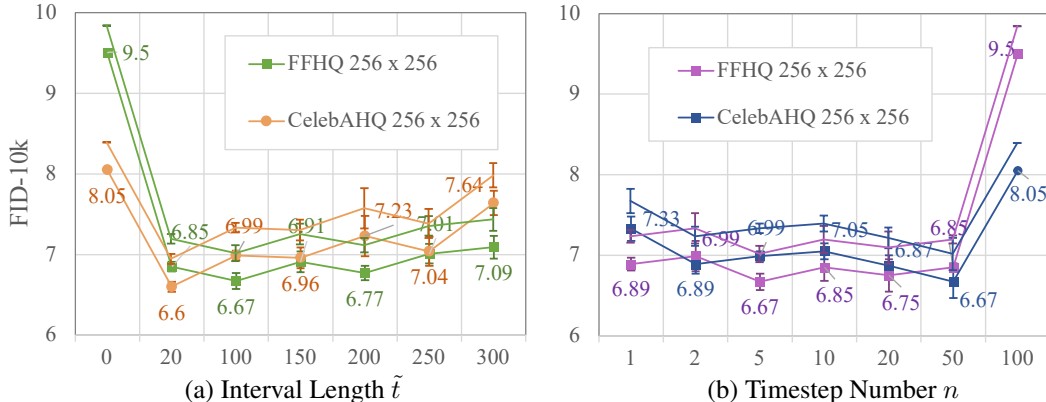

Figure 5: **Ablation study** on the length of the interval $t \in [0, \tilde{t})$ to share the timestep conditions, $\tilde{t}$, and the number of sub-intervals in this interval, $n$, using FID-10k $\downarrow$ as the evaluation metric. We repeat each experiment three times and provide the error bars.

Table 3: **Quantitative comparison** on FFHQ $256 \times 256$ (Karras et al., 2019) between DDPM (Ho et al., 2020) and our proposed E-TSDM utilizing different fast samplers, DDIM (Song et al., 2020) and DPM-Solver (Lu et al., 2022b), varying the number of function evalutaions (NFE). FID-10k$\downarrow$ is used as the evaluation metric, and DPM-Solver-$k$ represents the $k$-th-order DPM-Solver. Results indicate that **our approach well supports the popular fast samplers**.

| Fast Samplers | | DPM-Solver-3 | | DPM-Solver-2 | | DDIM | |
|---|---|---|---|---|---|---|---|
| NFE | | 20 | 50 | 20 | 50 | 50 | 200 |
| Method | DDPM | 21.91 | 24.48 | 22.21 | 24.80 | 21.80 | 23.16 |
| | E-TSDM | **16.97** | **13.52** | **17.28** | **14.14** | **19.34** | **13.71** |

a continuous-time version of E-TSDM. Specifically, we explore the three popular ones including linear, quadratic and cosine schedules (Nichol & Dhariwal, 2021), and two newly proposed ones, which are cosine-shift (Hoogeboom et al., 2023) and zero-terminal-SNR (Lin et al., 2023) schedules.

As shown in Table 2, our E-TSDM achieves excellent performance across different noise schedules. Besides, the comparison of Lipschitz constants between E-TSDM and baseline on different noise schedules, as illustrated in Appendix D.1, show that E-TSDM can mitigate the Lipschitz singularities issue besides the scenario of the linear schedule, highlighting that its effects are independent of the specific noise schedule. Additionally, the continuous-time version of E-TSDM outperforms the corresponding baseline, indicating that E-TSDM is effective for both continuous-time and discrete-time diffusion models. We provide the curves of the Lipschitz constants $K(t, t')$ in Figure A1 to compare continuous-time E-TSDM with its baseline on the linear schedule, showing that E-TSDM can mitigate Lipschitz singularities in the continuous-time scenario.

### 5.3.2 QUANTITATIVE ANALYSIS ON $n$ AND $\tilde{t}$

E-TSDM involves dividing the target interval $t \in [0, \tilde{t})$ with large Lipschitz constants into $n$ sub-intervals and sharing timestep conditions within each sub-interval. Accordingly, the choices of $\tilde{t}$ and $n$ have significant impacts on the performance of E-TSDM. Intuitively, $\tilde{t}$ should be a relatively small value, therefore representing an interval near zero point. As for $n$, it should not be too large or too small. If $n$ is too small, it forces the network to adapt to too many noise levels with a single timestep condition, thus leading to inaccuracy. Conversely, if the value of $n$ is set too large, the reduction of Lipschitz constants is insufficient, where the extreme situation is baseline.

In this section, we meticulously assess the impacts of $\tilde{t}$ and $n$ on various datasets. We present the outcomes on FFHQ $256 \times 256$ (Karras et al., 2019) and CelebAHQ $256 \times 256$ (Karras et al., 2017) for each hyperparameter in Figure 5, while leaving the remaining results in Appendix D.2. Specifically, in the experiments of $\tilde{t}$, we maintain the length of each sub-interval, namely, $\tilde{t}/n$, unchanged, while in the experiments of $n$, we maintain the $\tilde{t}$ unchanged. The results for $\tilde{t}$ in Figure 5 a demonstrate that E-TSDM performs well when $\tilde{t}$ is relatively small. However, as $\tilde{t}$ increases, the performance of E-TSDM deteriorates gradually. Furthermore, the results for $n$ are shown in Figure 5 b, from

which we observe a rise in FID when $n$ was too small, for instance, when $n = 2$. Conversely, when $n$ is too large, such as $n = 100$, the performance deteriorates significantly. Although E-TSDM performs well for most $n$ and $\tilde{t}$ values, considering the results on all of the evaluated datasets (see remaining results in Appendix D.2), $n = 5$ and $\tilde{t} = 100$ are recommended to be good choices to avoid cumbersome searches or a good starting point for further exploration when applying E-TSDM.

## 5.4 FAST SAMPLING

With the development of fast sampling algorithms, it is crucial that E-TSDM can be effectively combined with classic fast samplers, such as DDIM (Song et al., 2020) and DPM-Solver (Lu et al., 2022b). To this end, we incorporate both DDIM (Song et al., 2020) and DPM-Solver (Lu et al., 2022b) into E-TSDM for fast sampling in this section. It is worth noting that the presence of large Lipschitz constants can have a more detrimental impact on the efficiency of fast sampling compared to full-timestep sampling, as numerical solvers typically depend on the similarity between function values and their derivatives on adjacent steps. When using fast sampling algorithms with larger discretization steps, it becomes necessary for the functions to exhibit better smoothness, which in turn corresponds to smaller Lipschitz constants. Hence, it is anticipated that the utilization of E-TSDM will lead to an improvement in the generation performance of fast sampling methods.

As presented in Table 3, we observe that E-TSDM significantly outperforms the baseline when using the same number of function evaluations (NFE) for fast sampling, which is under expectation. Besides, the advantage of E-TSDM becomes more pronounced when using higher order sampler (from DDIM to DPM-Solver), indicating better smoothness when compared to the baseline. Notably, for both DDIM and DPM-Solver, we observe an abnormal phenomenon for baseline, whereby the performance deteriorates as NFE increases. This phenomenon has been previously noted by several works (Lu et al., 2022b;c; Li et al., 2023), but remains unexplained. Given that this phenomenon is not observed in E-TSDM, we hypothesize that it may be related to the improvement of smoothness of the learned network. We leave further verification of this hypothesis for future work.

## 5.5 CONDITIONAL GENERATION

In order to explore the potential for extending E-TSDM to conditional generation tasks, we further investigate its performance in the super-resolution task, which is one of the most popular conditional generation tasks. Specifically, we test E-TSDM on the FFHQ $256 \times 256$ dataset, using the $64 \times 64 \rightarrow 256 \times 256$ pixel resolution as our experimental settings. For the baseline in the super-resolution task, we utilize the same network structure and hyper-parameters as those employed in the baseline presented in Section 5.1, but incorporate a low-resolution image as an additional input. Besides, for E-TSDM, we adopt a general setting with $n = 5$ and $\tilde{t} = 100$. As illustrated in Figure A12, we observe that the baseline tends to exhibit a color bias compared to real images, which is mitigated by E-TSDM. Quantitatively, our results indicate that E-TSDM outperforms the baseline on the test set, achieving an improvement in PSNR from 24.64 to 25.61. These findings suggest that E-TSDM holds considerable promise for application in conditional generation tasks.

## 6 CONCLUSION

In this paper, we elaborate on the infinite Lipschitz of the diffusion model near the zero point from both theoretical and empirical perspectives, which hurts the stability and accuracy of the diffusion process. A novel E-TSDM is further proposed to mitigate the corresponding singularities in a timestep-sharing manner. Experimental results demonstrate the superiority of our method in both performance and adaptability to the baselines, including unconditional generation, conditional generation, and fast sampling. This paper may not only improve the performance of diffusion models, but also help to make up the critical research gap in the understanding of the rationality underlying the diffusion process.

**Limitations.** Although E-TSDM has demonstrated significant improvements in various applications, it has yet to be verified on large-scale text-to-image generative models. As E-TSDM reduces the large Lipschitz constants by sharing conditions, it is possible to lead to a decrease in the effectiveness of large-scale generative models. Additionally, the reduction of Lipschitz constants to zero within each sub-interval in E-TSDM may introduce unknown and potentially harmful effects.

## ACKNOWLEDGMENTS

We would like to thank the four anonymous reviewers for spending time and effort and bringing in constructive questions and suggestions, which helped us greatly to improve the quality of the paper. We would like to also thank the Program Chairs and Area Chairs for handling this paper and providing valuable and comprehensive comments. In addition, this research was funded by the Alibaba Innovative Research (AIR) project.

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

APPENDIX

## A  OVERVIEW

This supplementary material is organized as follows. First, to facilitate the reproducibility of our experiments, we present implementation details, including hyper-parameters in Appendix B.1 and algorithmic details in Appendix B.2. Next, in Appendix C, we provide all details of deduction involved in the main paper. Finally, we present additional experimental results in support of the effectiveness of E-TSDM.

## B  IMPLEMENTATION DETAILS

### B.1  HYPER-PARAMETERS

The hyper-parameters used in our experiments are shown in Table A1, and we use identical hyper-parameters for all evaluated datasets for both E-TSDM and their corresponding baselines. Specifically, we follow the hyper-parameters of DDPM (Ho et al., 2020) but adopt a more advanced structure of U-Net (Dhariwal & Nichol, 2021) with residual blocks from BigGAN (Brock et al., 2018). The network employs a block consisting of fully connected layers to encode the timestep, where the dimensionality of hidden layers for this block is determined by the timestep channels shown in Table A1. Moreover, we scale up the network to achieve the state-of-the-art results of diffusion

Table A1: **Hyper-parameters** of E-TSDM and our reproduced baseline.

|  | Normal version | Large version |
|---|---|---|
| $T$ | 1,000 | 1,000 |
| $\beta_t$ | linear | linear |
| Model size | 131M | 692M |
| Base channels | 128 | 128 |
| Channels multiple | (1,1,2,2,4,4) | (1,1,2,4,6,8) |
| Heads channels | 64 | 64 |
| Self attention | 32,16,8 | 32,64,8 |
| Timestep channels | 512 | 2048 |
| BigGAN block | ✓ | ✓ |
| Dropout | 0.0 | 0.0 |
| Learning rate | $1e^{-4}$ | $1e^{-4}$ |
| Batch size | 96 | 64 |
| Res blocks | 2 | 4 |
| EMA | 0.9999 | 0.9999 |
| Warmup steps | 0 | 0 |
| Gradient clip | ✗ | ✗ |

models on FFHQ $256 \times 256$ (Karras et al., 2019), and therefore we provide the hyper-parameters of the large version of E-TSDM in Table A1.

### B.2  ALGORITHM DETAILS

In this section, we provide a detailed description of the E-TSDM algorithm, including the training and inference procedures as shown in Algorithm A1 and Algorithm A2, respectively. Our method is simple to implement and requires only a few steps. Firstly, a suitable length of the interval $\tilde{t}$ should be selected for sharing conditions, along with the corresponding number of timestep conditions $n$ in the target interval $t \in [0, \tilde{t})$. While performing a thorough search for different datasets can achieve better performance, the default settings $\tilde{t} = 100$ and $n = 5$ are recommended when E-TSDM is applied without a thorough search.

Next, the target interval $t \in [0, \tilde{t})$ should be divided into $n$ sub-intervals, and the boundaries for each sub-interval should be calculated to generate the partition schedule $\mathbb{T} = \{t_0, t_1, \ldots, t_n\}$. Finally,

---

**Algorithm A1** Training of E-TSDM

---

**Require:** The length of the target interval $\tilde{t}$.
**Require:** The number of conditions $n$.
**Require:** Model $\epsilon_\theta$ to be trained.
**Require:** Data set $\mathcal{D}$.
 1: Uniformly divide the target interval $t \in [0, \tilde{t})$ into $n$ sub-intervals to get the corresponding timestep partition schedule $\mathbb{T} = \{t_0, t_1, \ldots, t_n\}$.
 2: **repeat**
 3:     $\mathbf{x}_0 \sim \mathcal{D}$
 4:     $t \sim \text{Uniform}(\{1, \ldots, T\})$
 5:     **if** $t < \tilde{t}$ **then**
 6:         $\hat{t} = \max_{1 \leq i \leq n}\{t_{i-1} \in \mathbb{T} : t_{i-1} \leq t\}$
 7:     **else**
 8:         $\hat{t} = t$
 9:     **end if**
10:     $\epsilon \sim \mathcal{N}(\mathbf{0}, \mathbf{I})$
11:     Take gradient descent step on
12:         $\nabla_\theta \| \epsilon - \epsilon_\theta(\alpha_t \mathbf{x}_0 + \sigma_t \epsilon, \hat{t}) \|^2$
13: **until** converged

---

**Algorithm A2** Sampling of E-TSDM

---

**Require:** The length of the target interval $\tilde{t}$.
**Require:** The number of conditions $n$.
**Require:** A trained model $\epsilon_\theta$.
 1: Uniformly divide the target interval $t \in [0, \tilde{t})$ into $n$ sub-intervals to get the corresponding timestep partition schedule $\mathbb{T} = \{t_0, t_1, \ldots, t_n\}$.
 2: $\mathbf{x}_T \sim \mathcal{N}(\mathbf{0}, \mathbf{I})$
 3: **for** $t = T, \ldots, 1$ **do**
 4:     **if** $t < \tilde{t}$ **then**
 5:         $\hat{t} = \max_{1 \leq i \leq n}\{t_{i-1} \in \mathbb{T} : t_{i-1} \leq t\}$
 6:     **else**
 7:         $\hat{t} = t$
 8:     **end if**
 9:     **if** $t > 1$ **then**
10:         $\mathbf{z} \sim \mathcal{N}(\mathbf{0}, \mathbf{I})$
11:     **else**
12:         $\mathbf{z} = 0$
13:     **end if**
14:     $\mathbf{x}_{t-1} = \frac{\alpha_{t-1}}{\alpha_t}\left(\mathbf{x}_t - \frac{\beta_t}{\sigma_t}\epsilon_\theta(\mathbf{x}_t, \hat{t})\right) + \eta_t \mathbf{z}$
15: **end for**
16: **return** $\mathbf{x}_0$

---

during both training and sampling, the corresponding left boundary $\hat{t}$ for each timestep in the target interval $t \in [0, \tilde{t})$ should be determined according to $\mathbb{T}$, and used as the conditional input of the network instead of $t$.

## C  DERIVATION OF FORMULAS

In this section, we provide detailed derivations as a supplement to the main paper. The derivations are divided into three parts, firstly we prove that the key assumption of the occurrence of Lipschitz singularities, $\frac{d\alpha_t}{dt}\big|_{t=0} \neq 0$, holds for mainstream noise schedules including linear, quadratic, and cosine schedules. Therefore, all of the diffusion models utilizing these noise schedules suffer from the issue of Lipschitz singularities. Then we show that Lipschitz singularities also plague the v-prediction (Salimans & Ho, 2022) models. Considering that most of the diffusion models are noise-prediction or v-prediction models, the Lipschitz singularities problem is an important issue for the

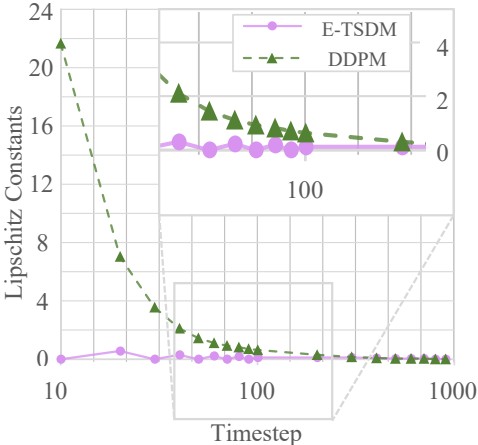

Figure A1: **Quantitative comparison** of the Lipschitz constants between **continuous-time** E-TSDM and **continuous-time** DDPM (Ho et al., 2020). Results show that E-TSDM can efficiently reduce the Lipschitz constants in **continuous-time** scenarios.

community of diffusion models. Finally, we demonstrate the detailed derivation of Theorem 4.1, showing that the errors introduced by E-TSDM can be bounded by an infinitesimal and thus are insignificant.

## C.1 $\mathrm{d}\alpha_t/\mathrm{d}t$ FOR WIDELY USED NOISE SCHEDULES AT ZERO POINT

We have already shown that for an arbitrary complex distribution, given a noise schedule, if $\frac{\mathrm{d}\alpha_t}{\mathrm{d}t}\big|_{t=0} \neq 0$, then we often have $\limsup_{t \to 0+} \left\| \frac{\partial \boldsymbol{\epsilon}_\theta(\mathbf{x},t)}{\partial t} \right\| \to \infty$, indicating the infinite Lipschitz constants around $t = 0$. In this section, we prove that $\frac{\mathrm{d}\alpha_t}{\mathrm{d}t}\big|_{t=0} \neq 0$ stands for three mainstream noise schedules including linear schedule, quadratic schedule and cosine schedule.

### C.1.1 $\mathrm{d}\alpha_t/\mathrm{d}t$ FOR LINEAR AND QUADRATIC SCHEDULES AT ZERO POINT

Linear and quadratic schedules are first proposed by Ho et al. (2020). Both of them determine $\{\alpha_t\}_{t=1}^T$ by a pre-designed positive sequence $\{\beta_t\}_{t=1}^T$ and the relationship $\alpha_t := \prod_{i=1}^t \sqrt{1 - \beta_i}$. Note that $t \in \{1, 2, \cdots, T\}$ is a discrete index, and $\{\alpha_t\}_{t=1}^T$, $\{\beta_t\}_{t=1}^T$ are discrete parameter sequences in DDPM. However, $\alpha_t$ in $\mathrm{d}\alpha_t/\mathrm{d}t$ refers to the continuous-time parameter determined by the following score SDE (Song et al., 2021b)

$$\mathrm{d}\mathbf{x}(\tau) = -\frac{1}{2}\beta(\tau)\mathbf{x}(\tau)\mathrm{d}\tau + \sqrt{\beta(\tau)}\mathrm{d}\mathbf{w}, \ \tau \in [0,1], \tag{A1}$$

where $\mathbf{w}$ is the standard Wiener process, $\beta(\tau)$ is the continuous version of $\{\beta_t\}_{t=1}^T$ with a continuous time variable $\tau \in [0,1]$ for indexing, and the continuous-time $\alpha_t = \exp\left(-\frac{1}{2}\int_0^t \beta(s)\mathrm{d}s\right)$. To avoid ambiguity, let $\alpha(\tau)$, $\tau \in [0,1]$ denote the continuous version of $\{\alpha_t\}_{t=1}^T$. Thus,

$$\frac{\mathrm{d}\alpha(\tau)}{\mathrm{d}\tau}\bigg|_{\tau=0} = -\frac{1}{2}\beta(\tau)\exp\left(-\frac{1}{2}\int_0^\tau \beta(s)\mathrm{d}s\right)\bigg|_{\tau=0} = -\frac{1}{2}\beta(0). \tag{A2}$$

Once the continuous function $\beta(\tau)$ is determined for a specific noise schedule, we can obtain $\frac{\mathrm{d}\alpha(\tau)}{\mathrm{d}\tau}\big|_{\tau=0}$ immediately by Equation (A2).

To obtain $\beta(\tau)$, we first give the expression of $\{\beta_t\}_{t=1}^T$ in linear and quadratic schedules (Ho et al., 2020)

$$\text{Linear:} \ \beta_t = \frac{\bar{\beta}_{\min}}{T} + \left(\frac{\bar{\beta}_{\max}}{T} - \frac{\bar{\beta}_{\min}}{T}\right) \cdot \frac{t-1}{T-1}, \tag{A3}$$

$$\text{Quadratic:} \ \beta_t = \left(\sqrt{\frac{\bar{\beta}_{\min}}{T}} + \left(\sqrt{\frac{\bar{\beta}_{\max}}{T}} - \sqrt{\frac{\bar{\beta}_{\min}}{T}}\right) \cdot \frac{t-1}{T-1}\right)^2, \tag{A4}$$

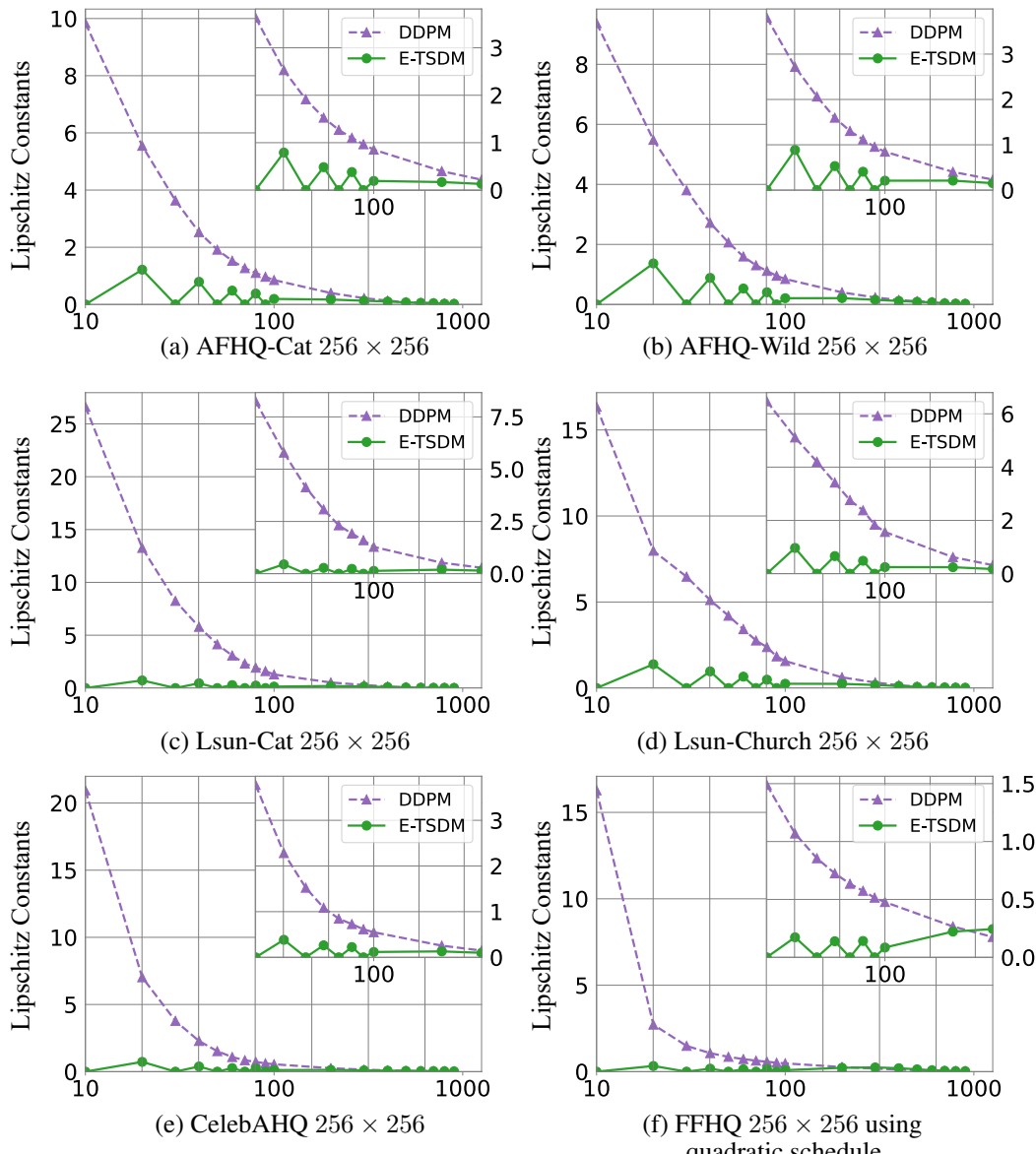

Figure A2: **Quantitative comparison** of Lipschitz constants between E-TSDM and DDPM baseline (Ho et al., 2020) **on various datasets**, including (a) AFHQ-Cat (Choi et al., 2020), (b) AFHQ-Wild (Choi et al., 2020), (c) Lsun-Cat $256 \times 256$ (Karras et al., 2019), (d) Lsun-Church $256 \times 256$ (Karras et al., 2019), and (e) CelebAHQ $256 \times 256$ (Karras et al., 2017) using the linear schedule. (f) **Quantitative comparison** of Lipschitz constants between E-TSDM and DDPM baseline (Ho et al., 2020) on FFHQ $256 \times 256$ (Karras et al., 2019) using the **quadratic schedule**.

where $\bar{\beta}_{\min}$ and $\bar{\beta}_{\max}$ are user-defined hyperparameters. Then, we define an auxiliary sequence $\{\bar{\beta}_t = T\beta_t\}_{t=1}^{T}$. In the limit of $T \to \infty$, $\{\bar{\beta}_t\}_{t=1}^{T}$ becomes the function $\beta(\tau)$ indexed by $\tau \in [0, 1]$

$$\text{Linear: } \beta(\tau) = \bar{\beta}_{\min} + \left(\bar{\beta}_{\max} - \bar{\beta}_{\min}\right) \cdot \tau, \tag{A5}$$

$$\text{Quadratic: } \beta(\tau) = \left(\sqrt{\bar{\beta}_{\min}} + \left(\sqrt{\bar{\beta}_{\max}} - \sqrt{\bar{\beta}_{\min}}\right) \cdot \tau\right)^2, \tag{A6}$$

Thus, $\beta(0) = \bar{\beta}_{\min}$ for both linear and quadratic schedules, which leads to $\left.\frac{\mathrm{d}\alpha(\tau)}{\mathrm{d}\tau}\right|_{\tau=0} = -\frac{1}{2}\bar{\beta}_{\min}$. As a common setting, $\bar{\beta}_{\min}$ is a positive real number, thus $\left.\frac{\mathrm{d}\alpha(\tau)}{\mathrm{d}\tau}\right|_{\tau=0} < 0$.

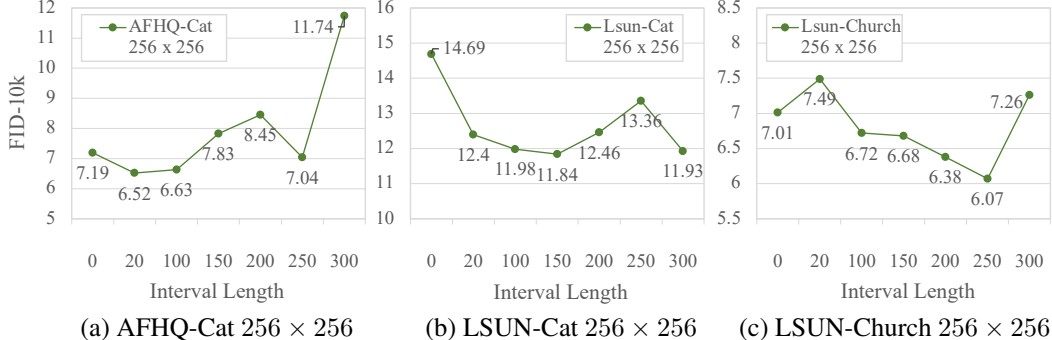

Figure A3: **Ablation study** on the length of the interval $t \in [0, \tilde{t})$ to share the timestep conditions, $\tilde{t}$, using FID-10k $\downarrow$ as the evaluation metric.

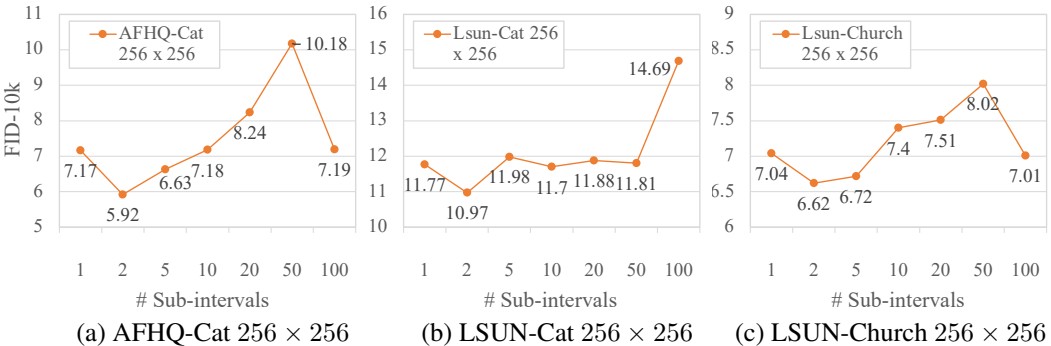

Figure A4: **Ablation study** on the number of sub-intervals in this interval, $n$, using FID-10k $\downarrow$ as the evaluation metric.

### C.1.2 $\mathrm{d}\alpha_t/\mathrm{d}t$ FOR THE COSINE SCHEDULE AT ZERO POINT

The cosine schedule is designed to prevent abrupt changes in noise level near $t = 0$ and $t = T$ (Nichol & Dhariwal, 2021). Different from linear and quadratic schedules that define $\{\alpha_t\}_{t=1}^T$ by a pre-designed sequence $\{\beta_t\}_{t=1}^T$, the cosine schedule directly defines $\{\alpha_t\}_{t=1}^T$ as

$$\alpha_t = \frac{f(t)}{f(0)}, \quad f(t) = \cos\left(\frac{t/T + s}{1 + s} \cdot \frac{\pi}{2}\right), \quad t = 1, 2, \cdots, T, \tag{A7}$$

where $s$ is a small positive offset. The continuous version of $\{\alpha_t\}_{t=1}^T$ can be obtained in the limit of $T \to \infty$ as

$$\alpha(\tau) = \cos\left(\frac{\tau + s}{1 + s} \cdot \frac{\pi}{2}\right) / \cos\left(\frac{s}{1 + s} \cdot \frac{\pi}{2}\right), \quad \tau \in [0, 1]. \tag{A8}$$

With Equation (A8), we can easily get $\left.\frac{\mathrm{d}\alpha(\tau)}{\mathrm{d}\tau}\right|_{\tau=0}$

$$\left.\frac{\mathrm{d}\alpha(\tau)}{\mathrm{d}\tau}\right|_{\tau=0} = -\frac{\pi}{2(1 + s)} \tan\left(\frac{s}{1 + s} \cdot \frac{\pi}{2}\right), \tag{A9}$$

which leads to $\left.\frac{\mathrm{d}\alpha(\tau)}{\mathrm{d}\tau}\right|_{\tau=0} < 0$ since $s > 0$.

### C.2 LIPSCHITZ SINGULARIES FOR V-PREDICTION DIFFUSION MODELS

In Section 3 of the main paper, we prove that noise-prediction diffusion models suffer from Lipschitz singularities issue. In this section, we show that the Lipschitz singularities issue is also an important problem for v-prediction diffusion models from both theoretical and empirical perspectives.

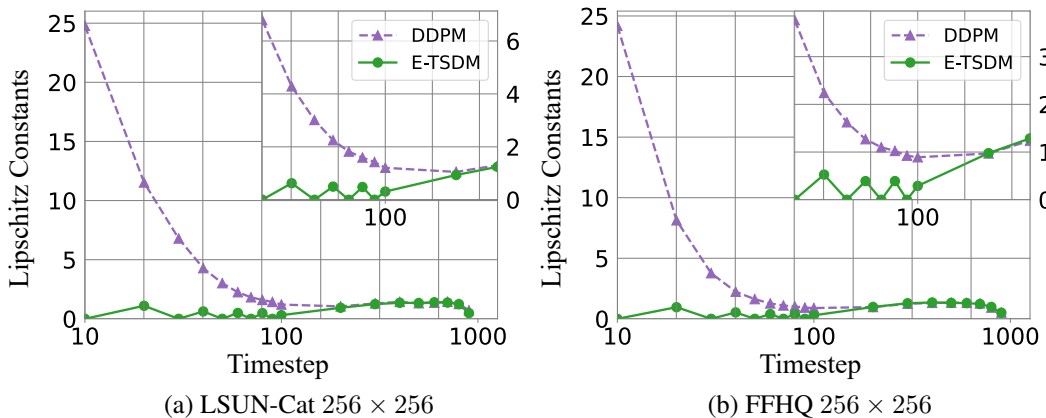

(a) LSUN-Cat $256 \times 256$             (b) FFHQ $256 \times 256$

Figure A5: **Quantitative comparison** of the Lipschitz constants between E-TSDM and DDPM (Ho et al., 2020) using **v-prediction** (Salimans & Ho, 2022) on Lsun-Cat $256 \times 256$ (Karras et al., 2019) and FFHQ $256 \times 256$ dataset (Karras et al., 2019). Results show that E-TSDM can efficiently reduce the Lipschitz constants in **v-prediction** scenarios.

Theoretically, the optimal solution of v-prediction models is

$$v(\mathbf{x}, t) = \underset{v_\theta}{\operatorname{argmin}} \mathbb{E}[\|v_\theta(\mathbf{x_t}, t) - (\alpha_t \boldsymbol{\epsilon} - \sigma_t \mathbf{x}_0)\|_2^2 | \mathbf{x}_t = \mathbf{x}]$$

$$= \mathbb{E}[\alpha_t \boldsymbol{\epsilon} - \sigma_t \mathbf{x}_0 | \mathbf{x}_t = \mathbf{x}]$$

$$= \mathbb{E}\left[\alpha_t \boldsymbol{\epsilon} - \sigma_t \frac{\mathbf{x}_t - \sigma_t \boldsymbol{\epsilon}}{\alpha_t} \middle| \mathbf{x}_t = \mathbf{x}\right] \tag{A10}$$

$$= -\frac{\sigma_t}{\alpha_t} x + (\alpha_t + \frac{\sigma_t^2}{\alpha_t}) \mathbb{E}[\boldsymbol{\epsilon} | \mathbf{x}_t = \mathbf{x}]$$

$$= -\frac{\sigma_t}{\alpha_t} x - \frac{\alpha_t^2 + \sigma_t^2}{\alpha_t} \sigma_t \nabla_x \log q_t(\mathbf{x})$$

$$= -\frac{\sigma_t}{\alpha_t} (\mathbf{x} + \nabla_\mathbf{x} \log q_t(\mathbf{x})),$$

where $\mathbf{x} + \nabla_\mathbf{x} \log q_t(\mathbf{x})$ is smooth under the assumption of Theorem 3.1, and $\frac{\mathrm{d}}{\mathrm{d}t}\left(\frac{\sigma_t}{\alpha_t}\right) \to \frac{\mathrm{d}\sigma_t}{\mathrm{d}t}$ as $t \to 0$. Thus, with the same derivation of Theorem 3.1, we can conclude that $\limsup_{t \to 0^+} \left\|\frac{\partial v(x,t)}{\partial t}\right\| \to \infty$. The detailed derivation goes as follows:

Firstly, we can obtain the partial derivative of the v-prediction model over $t$ as

$$\frac{\partial v(\mathbf{x}, t)}{\partial t} = -\frac{\mathrm{d}}{\mathrm{d}t}(\frac{\sigma_t}{\alpha_t})(\mathbf{x} + \nabla_\mathbf{x} \log q_t(\mathbf{x})) - \frac{\sigma_t}{\alpha_t} \frac{\mathrm{d}}{\mathrm{d}t}(\mathbf{x} + \nabla_\mathbf{x} \log q_t(\mathbf{x})). \tag{A11}$$

Note that $\frac{\mathrm{d}}{\mathrm{d}t}\left(\frac{\sigma_t}{\alpha_t}\right) = \frac{1}{\alpha_t^2}\left(\alpha_t \frac{\mathrm{d}\sigma_t}{\mathrm{d}t} - \sigma_t \frac{\mathrm{d}\alpha_t}{\mathrm{d}t}\right) \to \frac{\mathrm{d}\sigma_t}{\mathrm{d}t} = -\frac{\alpha_t}{\sqrt{1-\alpha_t^2}} \frac{\mathrm{d}\alpha_t}{\mathrm{d}t}$ as $t \to 0$ under common settings that $\sigma_0 = 0$, $\alpha_0 = 1$, and $\frac{\mathrm{d}\alpha_t}{\mathrm{d}t}\big|_{t=0}$ is finite, thus if $\frac{\mathrm{d}\alpha_t}{\mathrm{d}t}\big|_{t=0} \neq 0$, and $\mathbf{x} + \nabla_\mathbf{x} \log q_t(\mathbf{x}) \neq \mathbf{0}$, then one of the following two must stand

$$\limsup_{t \to 0^+} \left\|\frac{\partial v(\mathbf{x}, t)}{\partial t}\right\| \to \infty; \quad \limsup_{t \to 0^+} \left\|\frac{\sigma_t}{\alpha_t} \frac{\mathrm{d}}{\mathrm{d}t}(\mathbf{x} + \nabla_\mathbf{x} \log q_t(\mathbf{x}))\right\| \to \infty. \tag{A12}$$

Under the assumption that $q_t(\mathbf{x})$ is a smooth process, we can conclude that $\limsup_{t \to 0^+} \left\|\frac{\partial v(\mathbf{x},t)}{\partial t}\right\| \to \infty$.

Table A2: **Quantitative comparison** between E-TSDM and DDPM (Ho et al., 2020) using **v-prediction** on Lsun-Cat $256 \times 256$ (Karras et al., 2019) and FFHQ $256 \times 256$ dataset (Karras et al., 2019) evaluated with FID-10k ↓. Experimental results indicate that E-TSDM can achieve better synthesis performance.

|  | Baseline | E-TSDM |
|---|---|---|
| FFHQ | 10.85 | **9.00** |
| Lsun-Cat | 18.40 | **13.86** |

Table A3: **Quantitative comparison** among E-TSDM, DDPM (Ho et al., 2020), and DDPM using **regularization** techniques (DDPM-r) on FFHQ $256 \times 256$ (Karras et al., 2019) and CelebAHQ $256 \times 256$ (Karras et al., 2017) evaluated with FID-10k ↓. Experimental results show that **DDPM-r can slightly improve the FID but performs worse than E-TSDM**.

| Method | Baseline | E-TSDM | DDPM-r |
|---|---|---|---|
| FFHQ | 9.50 | **6.62** | 9.18 |
| CelebAHQ | 8.05 | **6.99** | 7.97 |

Since most of the diffusion models are noise-prediction and v-prediction models, the Lipschitz singularities issue is an important problem for the community of diffusion models.

Empirically, we can also observe the phenomenon of Lipschitz singularities for v-prediction diffusion models, where the experimental results of Lipschitz constants on FFHQ $256 \times 256$ dataset (Karras et al., 2019) and Lsun-Cat $256 \times 256$ (Karras et al., 2019) are shown in Figure A5, from which we can tell E-TSDM can effectively mitigate Lipschitz singularities in v-prediction scenario. Besides, we also provide corresponding quantitative evaluations evaluated by FID-10k in Table A2, showing that E-TSDM can also improve the synthesis performance in the v-prediction scenario.

## C.3 PROOF OF THEOREM 4.1

Here we will first give the derivation of the upper-bound on $\|\boldsymbol{\epsilon}^*(\mathbf{x}, f_{\mathbb{T}}(t)) - \boldsymbol{\epsilon}(\mathbf{x}, t)\|$ when $t \in [0, \tilde{t})$, where $\boldsymbol{\epsilon}^*(\mathbf{x}, f_{\mathbb{T}}(t))$ denotes the optimal $\boldsymbol{\epsilon}_\theta(\mathbf{x}, f_{\mathbb{T}}(t))$, and $\boldsymbol{\epsilon}(\mathbf{x}, t) = -\sigma_t \nabla_\mathbf{x} \log q_t(\mathbf{x})$. Then, we will discuss the convergence rate of the error bound.

For any $t \in [0, \tilde{t})$, there exists an $i \in \{1, 2, \cdots, n\}$ such that $t \in [t_{i-1}, t_i)$. For simplicity, we use $h(\mathbf{x}, t)$ to denote the score function $\nabla_\mathbf{x} \log q_t(\mathbf{x})$, and use $\mathbb{E}_\tau[\cdot]$ to denote the expectation over $\tau \sim \mathcal{U}(t_{i-1}, t_i)$. Thus, we can obtain

$$
\begin{aligned}
\|\boldsymbol{\epsilon}^*(\mathbf{x}, f(t)) - \boldsymbol{\epsilon}(\mathbf{x}, t)\| &= \|\mathbb{E}_\tau[\boldsymbol{\epsilon}(\mathbf{x}, \tau)] - \boldsymbol{\epsilon}(\mathbf{x}, t)\| \\
&= \|\mathbb{E}_\tau[\sigma_\tau h(\mathbf{x}, \tau)] - \sigma_t h(\mathbf{x}, t)\| \\
&= \|\mathbb{E}_\tau[\sigma_\tau h(\mathbf{x}, \tau) - \sigma_\tau h(\mathbf{x}, t) + \sigma_\tau h(\mathbf{x}, t) - \sigma_t h(\mathbf{x}, t)]\| \\
&\leq \|\mathbb{E}_\tau[\sigma_\tau (h(\mathbf{x}, \tau) - h(\mathbf{x}, t))]\| + \|\mathbb{E}_\tau[(\sigma_\tau - \sigma_t)h(\mathbf{x}, t)]\| \\
&\leq \mathbb{E}_\tau[\sigma_\tau \|h(\mathbf{x}, \tau) - h(\mathbf{x}, t)\|] + \mathbb{E}_\tau[|\sigma_\tau - \sigma_t|]\|h(\mathbf{x}, t)\| \\
&\leq \sigma_{t_i} \mathbb{E}_\tau[\|h(\mathbf{x}, \tau) - h(\mathbf{x}, t)\|] + (\sigma_{t_i} - \sigma_{t_{i-1}})\|h(\mathbf{x}, t)\| \\
&\leq \sigma_{t_i} K_i(\mathbf{x})(t_i - t_{i-1}) + B_i(\mathbf{x})(\sigma_{t_i} - \sigma_{t_{i-1}}) \\
&\leq \sigma_{\tilde{t}} K(\mathbf{x})\Delta t + B(\mathbf{x})\Delta\sigma_{\max},
\end{aligned}
\tag{A13}
$$

where $K_i(\mathbf{x}) = \sup_{t,\tau \in [t_{i-1}, t_i), t \neq \tau} \frac{\|h(\mathbf{x},t) - h(\mathbf{x},\tau)\|}{|t-\tau|}$, $B_i(\mathbf{x}) = \sup_{t \in [t_{i-1}, t_i)} \|h(\mathbf{x}, t)\|$, $K(\mathbf{x}) = \sup_{t,\tau \in [0,\tilde{t}), t \neq \tau} \frac{\|h(\mathbf{x},t) - h(\mathbf{x},\tau)\|}{|t-\tau|}$, $B(\mathbf{x}) = \sup_{t \in [0,\tilde{t})} \|h(\mathbf{x}, t)\|$, and $\Delta\sigma_{\max} = \max_{1 \leq i \leq n} |\sigma_{t_i} - \sigma_{t_{i-1}}|$. The first equality holds because

$$
\begin{aligned}
\boldsymbol{\epsilon}(\mathbf{x}, t) &= \arg\min_{\boldsymbol{\epsilon}_\theta} \mathbb{E}[\|\boldsymbol{\epsilon}_\theta(\mathbf{x}_\tau, \tau) - \boldsymbol{\epsilon}\|_2^2 | \tau = t, \mathbf{x}_\tau = \mathbf{x}] \\
&= \mathbb{E}[\boldsymbol{\epsilon} | \tau = t, \mathbf{x}_\tau = \mathbf{x}],
\end{aligned}
\tag{A14}
$$

Table A4: **Quantitative comparison** among E-TSDM, DDPM (Ho et al., 2020), and **modification of noise schedules (Modified-NS)** on FFHQ $256 \times 256$ dataset (Karras et al., 2019) evaluated with FID-10k $\downarrow$. Specifically, we implement Modified-NS on linear, quadratic, and cosine schedules. Experimental results indicate that the performance of **Modified-NS is unstable while E-TSDM achieves better synthesis performance**.

| | Linear | Quadratic | Cosine |
|---|---|---|---|
| Baseline | 9.50 | 13.79 | 27.17 |
| E-TSDM | **6.62** | **9.69** | **26.08** |
| Modified-NS | 8.67 | 17.48 | 26.84 |

Table A5: **Quantitative comparison** of **remap** method between uniformly sampling $t$ and uniformly sampling $\lambda$, during training and inference, on FFHQ $256 \times 256$ (Karras et al., 2019) evaluated with FID-10k $\downarrow$. Specifically, $\mathcal{U}_t$ is $\mathcal{U}[0,1]$, and $\mathcal{U}_\lambda$ is $\mathcal{U}[0,K]$ for $1/t$ but $\mathcal{U}[-K,K]$ for Inverse-Sigmoid, where $K$ is a large number to avoid infinity. Results show that **remap is not helpful**.

| Training Strategy | Inference Strategy | Remap Function $1/t$ | Inverse-Sigmoid |
|---|---|---|---|
| $t \sim \mathcal{U}_t$ | $t \sim \mathcal{U}_t$ | 9.43 | 9.33 |
| $t \sim \mathcal{U}_t$ | $\lambda \sim \mathcal{U}_\lambda$ | 83.71 | 468.90 |
| $\lambda \sim \mathcal{U}_\lambda$ | $t \sim \mathcal{U}_t$ | 83.44 | 468.19 |
| $\lambda \sim \mathcal{U}_\lambda$ | $\lambda \sim \mathcal{U}_\lambda$ | 171.06 | 351.89 |

and our optimal $\boldsymbol{\epsilon}^*(\mathbf{x}, f(t))$ can be expressed as

$$
\begin{aligned}
\boldsymbol{\epsilon}^*(\mathbf{x}, f(t)) &= \boldsymbol{\epsilon}^*(\mathbf{x}, t_{i-1}) \\
&= \arg\min_{\boldsymbol{\epsilon}_\theta} \mathbb{E}_{\tau \sim \mathcal{U}(t_{i-1}, t_i), \boldsymbol{\epsilon}}[\|\boldsymbol{\epsilon}_\theta(\mathbf{x}_\tau, t_{i-1}) - \boldsymbol{\epsilon}\|_2^2 | \mathbf{x}_\tau = \mathbf{x}] \\
&= \mathbb{E}_{\tau \sim \mathcal{U}(t_{i-1}, t_i), \boldsymbol{\epsilon}}[\boldsymbol{\epsilon} | \mathbf{x}_\tau = \mathbf{x}] \\
&= \mathbb{E}_{\tau \sim \mathcal{U}(t_{i-1}, t_i)} \mathbb{E}_{\boldsymbol{\epsilon}}[\boldsymbol{\epsilon} | \tau, \mathbf{x}_\tau = \mathbf{x}] \\
&= \mathbb{E}_{\tau \sim \mathcal{U}(t_{i-1}, t_i)}[\boldsymbol{\epsilon}(\mathbf{x}, \tau)].
\end{aligned}
\tag{A15}
$$

As for the rate of convergence, it is obvious from Equation (A13) that we only need to determine the convergence rate of $\Delta \sigma_{\max}$. Under common settings, $\sigma_t$ is monotonically decreasing and concave for $t \in [0, T]$, thus

$$
\Delta \sigma_{\max} = \max_{1 \le i \le n} |\sigma_{t_i} - \sigma_{t_{i-1}}| = \sigma_{t_1} - \sigma_{t_0} = \sigma_{\Delta t},
\tag{A16}
$$

where the last equality holds because $\sigma_{t_0} = \sigma_0 = 0$, and $t_1 = \tilde{t}/n = \Delta t$ as we uniformly divides $[0, \tilde{t}]$ into $n$ sub-intervals. Then, we can verify the convergence rate of $\Delta \sigma_{\max}$ as

$$
\begin{aligned}
\lim_{\Delta t \to 0} \frac{\Delta \sigma_{\max}}{\sqrt{\Delta t}} &= \lim_{\Delta t \to 0} \sqrt{\frac{\sigma_{\Delta t}^2}{\Delta t}} \\
&= \sqrt{\left.\frac{d\sigma_t^2}{dt}\right|_{t=0}} \\
&= \sqrt{\left.\frac{d(1 - \alpha_t^2)}{dt}\right|_{t=0}} \\
&= \sqrt{\left.-2\alpha_t \frac{d\alpha_t}{dt}\right|_{t=0}} \\
&= \sqrt{\left.-2\frac{d\alpha_t}{dt}\right|_{t=0}},
\end{aligned}
\tag{A17}
$$

where $\left.\frac{d\alpha_t}{dt}\right|_{t=0}$ is finite and $\left.\frac{d\alpha_t}{dt}\right|_{t=0} \le 0$. Thus, we can conclude that $\Delta \sigma_{\max}$ is at least $\frac{1}{2}$-order convergence with respect to $\Delta t$, and the error bound $\sigma_{\tilde{t}} K(\mathbf{x}) \Delta t + B(\mathbf{x}) \Delta \sigma_{\max}$ is also at least $\frac{1}{2}$-order convergence. This is a relatively fast convergence speed in optimization, and demonstrates that the introduced errors of E-TSDM are controllable.

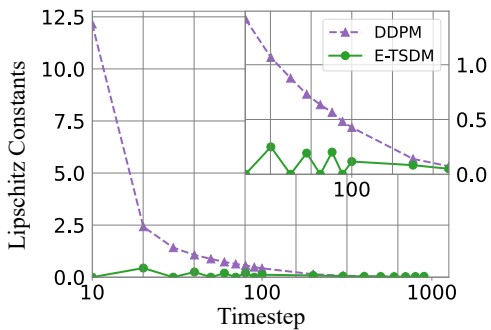
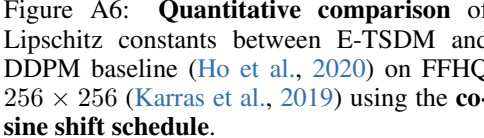
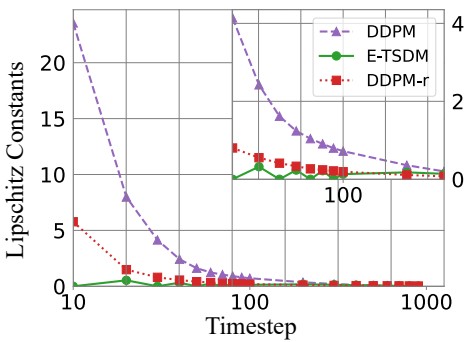

Figure A6: **Quantitative comparison** of Lipschitz constants between E-TSDM and DDPM baseline (Ho et al., 2020) on FFHQ $256 \times 256$ (Karras et al., 2019) using the **cosine shift schedule**.

Figure A7: **Quantitative comparison** of Lipschitz constants among E-TSDM, DDPM (Ho et al., 2020), and DDPM (Ho et al., 2020) using **regularization** techniques (DDPM-r) on FFHQ $256 \times 256$ (Karras et al., 2019).

# D    ADDITIONAL RESULTS

## D.1    LIPSCHITZ CONSTANTS

In our main paper, we demonstrate the effectiveness of E-TSDM in reducing the Lipschitz constants near $t = 0$ by comparing its Lipschitz constants with that of DDPM baseline (Ho et al., 2020) on the FFHQ $256 \times 256$ dataset (Karras et al., 2019). As a supplement, we provide additional comparisons of Lipschitz constants on other datasets, including AFHQ-Cat (Choi et al., 2020) (see Figure A2a), AFHQ-Wild (Choi et al., 2020) (see Figure A2b), Lsun-Cat $256 \times 256$ (Karras et al., 2019) (see Figure A2c), Lsun-Church $256 \times 256$ (Karras et al., 2019) (see Figure A2d), and CelebAHQ $256 \times 256$ (Karras et al., 2017) (see Figure A2e). These experimental results demonstrate that E-TSDM is highly effective in mitigating Lipschitz singularities in diffusion models across various datasets.

Furthermore, we provide a comparison of Lipschitz constants between E-TSDM and the DDPM baseline (Ho et al., 2020) when using the quadratic schedule and the cosine-shift schedule (Hoogeboom et al., 2023). As shown in Figure A2f, we observe that large Lipschitz constants still exist in diffusion models when using the quadratic schedule, and E-TSDM effectively alleviates this problem. Similar improvement can also be observed when using the cosine-shift schedule as illustrated in Figure A6, highlighting the superiority of our approach over the DDPM baseline.

## D.2    QUANTITATIVE ANALYSIS OF $\tilde{t}$ AND $n$

In our main paper, we investigated the impact of two important settings for E-TSDM, the length of the interval to share conditions $\tilde{t}$, and the number of sub-intervals $n$ in this interval. As a supplement, we provide additional results on various datasets to further investigate the optimal settings for these parameters.

As seen in Figure A3 and Figure A4, we observe divergence in the best choices of $n$ and $\tilde{t}$ across different datasets. However, we find that the default settings where $\tilde{t} = 100$ and $n = 5$ consistently yield good performance across a range of datasets. Based on these findings, we recommend the default settings as an ideal choice for implementing E-TSDM without the need for a thorough search. However, if performance is the main concern, researchers may conduct a grid search to explore the optimal values of $\tilde{t}$ and $n$ for specific datasets.

## D.3    ALTERNATIVE METHODS

In this section, we discuss three different alternative methods that possibly alleviate Lipschitz singularities. including regularization, modification of noise schedules, and remap. Although seem feasible, they have different problems, resulting in worse performance than E-TSDM.

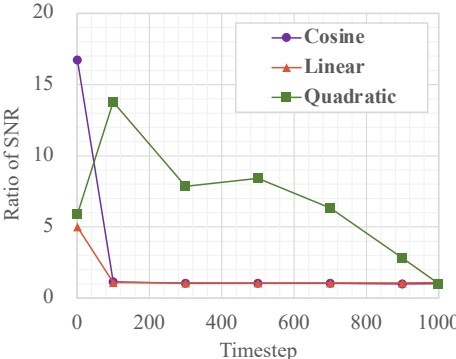 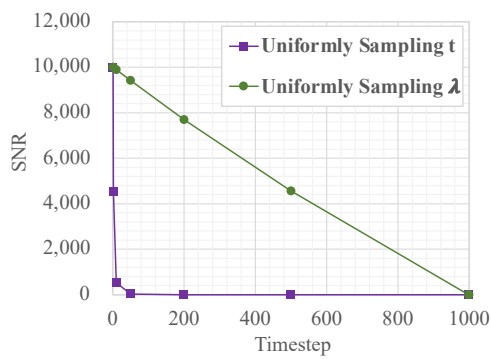

Figure A8: **Quantitative evaluation** of the **ratio of SNR** of Modified-NS to the SNR of the corresponding original noise schedule. Results show that **Modified-NS** significantly increases the SNR near zero point, and thus reduces the amounts of added noise near zero point. Specifically, for the quadratic schedule, Modified-NS seriously increases the SNR almost during the whole process.

Figure A9: **Quantitative comparison** of SNR for remap method between uniformly sampling $t$ and uniformly sampling the remapped conditional input $\lambda$. Results show that when using **remap** method, uniformly sampling $\lambda$ significantly increases the SNR across all of the timesteps, and thus forces the network to focus too much on the beginning stage of the diffusion process.

### D.3.1 REGULARIZATION

As mentioned in the main paper, one alternative method is to impose restrictions on the Lipschitz constants through regularization techniques. In this section, we apply regularization on the baseline and estimate the gradient of $\epsilon_\theta(\mathbf{x}, t)$ by calculating the difference $K(t, t')$. We represent this method as DDPM-r in this paper. As shown in Figure A7, although DDPM-r can also reduce the Lipschitz constants, its capacity to do so is substantially inferior to that of E-TSDM. Additionally, DDPM-r necessitates twice the calculation compared to E-TSDM. Regarding synthesis performance, as shown in Table A3, DDPM-r performs slightly better than baseline, but much worse than E-TSDM, indicating that E-TSDM is a better choice than regularization.

### D.3.2 MODIFYING NOISE SCHEDULES

As proved in Appendix C, the mainstream noise schedules satisfy $\frac{d\alpha_t}{dt}\big|_{t=0} \neq 0$, leading to Lipschitz singularities as proved in Theorem 3.1. However, it is possible to modify those schedules to force them to have $\frac{d\alpha_t}{dt}\big|_{t=0} = 0$, and thus alleviate Lipschitz singularities. We represent this method as Modified-NS in this paper. However, as said in Nichol & Dhariwal (2021), $\frac{d\alpha_t}{dt}\big|_{t=0} = 0$ means tiny amounts of noise at the beginning of the diffusion process, making it hard for the network to predict accurately enough.

To explore the performance, we conduct experiments of Modified-NS on FFHQ $256 \times 256$ (Karras et al., 2019) for all of the three discussed noise schedules in Appendix C.1. Specifically, for linear and quadratic schedules, since $\frac{d\alpha(\tau)}{d\tau}\big|_{\tau=0} = -\frac{1}{2}\beta(0)$ (as detailed in Equation (A2)), we implement Modified-NS by setting $\beta(0) = 0$. Note that for the quadratic schedule, such a modification will significantly magnify the Signal to Noise Ratio (SNR), $\frac{\alpha_t^2}{\sigma_t^2}$, across the whole diffusion process, so we slightly increase $\beta_T$ to make its SNR at $t = T$ similar to that of the original quadratic schedule. Meanwhile, $\beta_1, \ldots, \beta_{T-1}$ are also correspondingly increased due to $\beta_t = (\sqrt{\beta_0} + (\sqrt{\beta_T} - \sqrt{\beta_0})\frac{t}{T-1})^2$. As for the cosine schedule, we set the offset $s$ in Equation (A7) to zero. Experimental results are shown in Table A4, from which we find that the performance of Modified-NS is unstable. More specifically, Modified-NS improves performance for linear and cosine schedules but significantly drags down the performance for the quadratic schedule. We further provide the comparison of SNR between Modified-NS and their corresponding original noise schedules in Figure A8 by calculating the ratio of Modified-NS's SNR to the original noise schedule's SNR. From this figure we can tell that for linear and cosine schedule, Modified-NS significantly increase the SNR near zero point while maintaining the SNR of other timesteps similar. In other

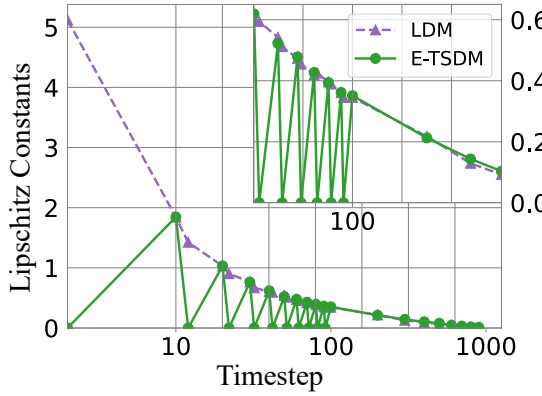

Figure A10: **Quantitative comparison** of Lipschitz constants between E-TSDM and LDM (Rombach et al., 2022) on FFHQ $256 \times 256$(Karras et al., 2019). E-TSDM reduces the overall Lipschitz constants near $t = 0$, and mitigates the Lipschitz singularities occurring in LDM (Rombach et al., 2022).

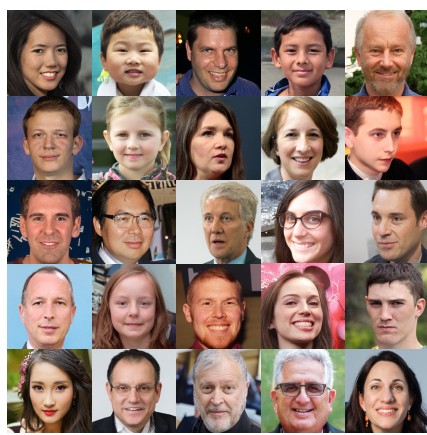

Figure A11: **Qualitative results** produced by E-TSDM implemented on LDM (Rombach et al., 2022) on FFHQ $256 \times 256$(Karras et al., 2019).

words, on the one hand, Modified-NS seriously reduces the amount of noise added near zero point, which can be detrimental to the accurate prediction. On the other hand, Modified-NS alleviates the Lipschitz singularities, which is beneficial to the synthesis performance. As a result, for linear and cosine schedules, Modified-NS performs better than baseline but worse than E-TSDM. However, for the quadratic schedule, although we force the SNR of Modified-NS at $t = T$ similar to the SNR of the original schedule, the SNR at other timesteps is significantly increased, leading to a worse performance of Modified-NS compared to that of baseline.

### D.3.3 REMAP

Except for regularization and Modified-NS, remap is another possible method to fix the Lipschitz singularities issue. Recall that the inputs of network $\epsilon_\theta(\mathbf{x}, t)$ is noisy image $\mathbf{x}$ and timestep condition $t$. Remap is trying to design a remap function $\lambda = f(t)$ on $t$ as the conditional input of network instead of $t$, namely, $\epsilon_\theta(\mathbf{x}, f(t))$. The core idea of remap is to reduce $\frac{\partial \epsilon_\theta(\mathbf{x},t)}{\partial t}$ by significantly stretching the interval with large Lipschitz constants. Note that although $f_{\mathbb{T}}$ of E-TSDM can also be seen as a kind of remap function, there are big differences between E-TSDM and remap. Specifically, E-TSDM tries to set the numerator to zero while remap aims to significantly increase the denominator. Besides, $f_{\mathbb{T}}$ has no inverse while $f(t)$ of remap is usually a reversible function. We provide two simple choices of $f(t)$ in this section as examples, which are $f(t) = 1/t$ and $f^{-1}(\lambda) = \text{sigmoid}(\lambda)$.

Remap can efficiently reduce the Lipschitz constants regarding the conditional inputs of the network, $\frac{\partial \epsilon_\theta(\mathbf{x},t)}{\partial \lambda}$. However, since we uniformly sample $t$ both in training and inference, what should be focused on is the Lipschitz constants regarding $t$, $\frac{\partial \epsilon_\theta(\mathbf{x},t)}{\partial t}$, which can not be influenced by remap. In other words, although remap seems to be a feasible method, it is not helpful to mitigate the Lipschitz constants we care about, unless we uniformly sample $\lambda$ in training and inference. However, uniformly sampling $\lambda$ may force the network to focus on a certain part of the diffusion process. We use $f(t) = 1/t$ as an example to illustrate this point and show the comparison of SNR between uniformly sampling $t$ and uniformly sampling $\lambda$ when using remap in Figure A9. Results show that uniformly sampling $\lambda$ maintains a high SNR across all of the timesteps, leading to excessive attention to the beginning stage of the diffusion process. As a result, when we uniformly sample $\lambda$ during training or inference, the synthesis performance gets significantly worse as shown in Table A5. Besides, when we uniformly sample $t$ both in training and inference, remap makes no difference and thus leads to a similar performance to the baseline.

Figure A12: **Qualitative and quantitative results** by applying E-TSDM to super-resolution task (*i.e.*, from $64 \times 64$ to $256 \times 256$), using PSNR as the evaluation metric. Results show that E-TSDM mitigates the color bias occurring in baseline and improves the PSNR from 24.64 to 25.61, which suggests that **our approach well supports conditional generation**.

## D.4 MORE DIFFUSION MODELS

Latent diffusion models (LDM) (Rombach et al., 2022) is one of the most renowned variants of diffusion models. In this section, we will investigate the Lipschitz singularities in LDM (Rombach et al., 2022), and apply E-TSDM to address this problem. LDM (Rombach et al., 2022) shares a resemblance with DDPM (Rombach et al., 2022) but has an additional auto-encoder to encode images into the latent space. As LDM typically employs the quadratic schedule, it is also susceptible to Lipschitz singularities, as confirmed in Figure A10.

As seen in Figure A10, by utilizing E-TSDM, the Lipschitz constants within each timestep-shared sub-interval are reduced to zero, while the timesteps located near the boundaries of the sub-intervals exhibit a Lipschitz constant comparable to that of baseline, leading to a decrease in overall Lipschitz constants in the target interval $t \in [0, \tilde{t})$, where $\tilde{t}$ is set as the default, namely $\tilde{t} = 100$. Consequently, E-TSDM achieves an improvement in FID-50k from 4.98 to 4.61 with the adoption of E-TSDM, when $n = 20$. We provide some samples generated by the E-TSDM implemented on LDM in Figure A11.

Besides, we also implement our E-TSDM to Elucidated diffusion models (EDM) (Karras et al., 2022), which proposed several changes to both the sampling and training processes and achieves impressive performance. Specifically, we reproduce EDM and repeat it three times on CIFAR10 $32 \times 32$ (Krizhevsky et al., 2009) to get a FID-50k of $1.904 \pm 0.015$, which is slightly worse than the official released one. Then we apply E-TSDM to EDM and repeat it three times to get a FID-50k of $1.797 \pm 0.016$, indicating that E-TSDM is also helpful to EDM.

## D.5 GENERATED SAMPLES

As a supplement, we provide massive generated samples of E-TSDM trained on Lsun-Church $256 \times 256$ (Karras et al., 2019) (see Figure A13), Lsun-Cat $256 \times 256$ (Karras et al., 2019) (see Figure A14), AFHQ-Cat $256 \times 256$ (Choi et al., 2020), AFHQ-Wild $256 \times 256$ (Choi et al., 2020) (see Figure A15), FFHQ $256 \times 256$ (Karras et al., 2019) (see Figure A16), and CelebAHQ $256 \times 256$ (Karras et al., 2017) (see Figure A17).

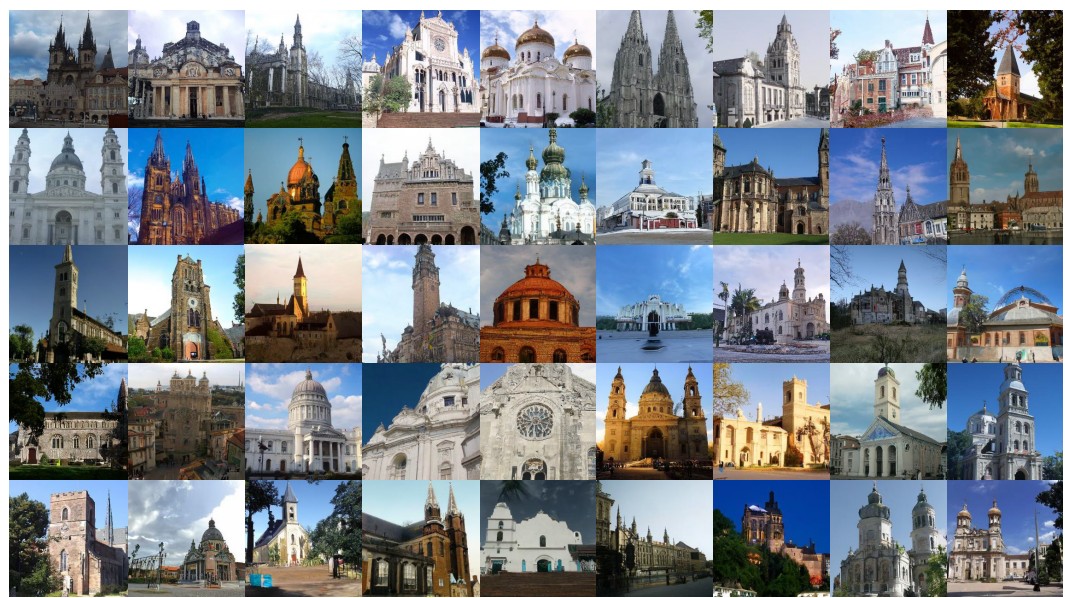

Figure A13: **Qualitative results** produced by E-TSDM on Lsun-Church $256 \times 256$ (Yu et al., 2015).

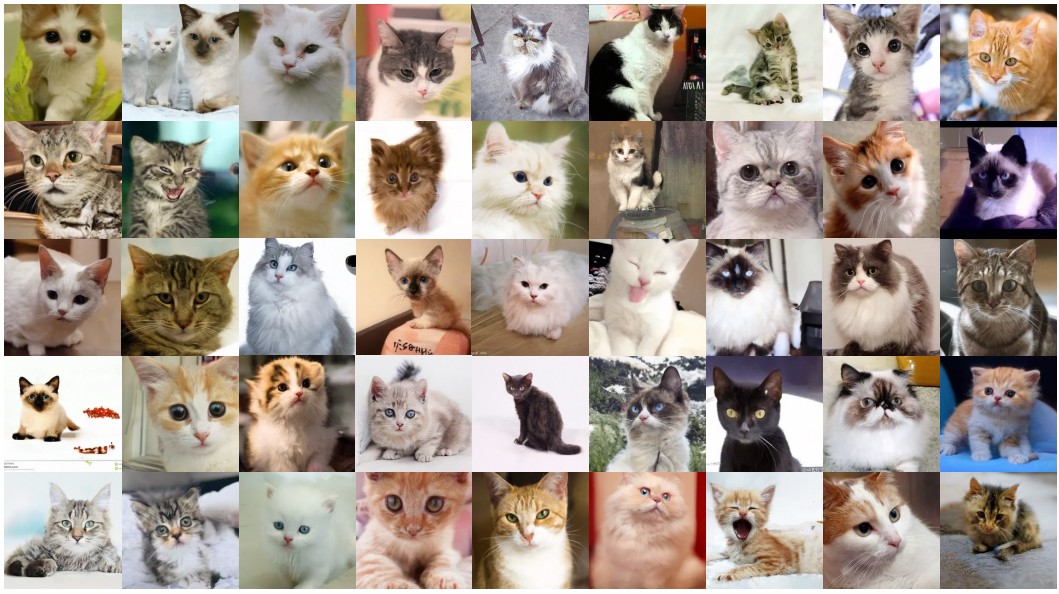

Figure A14: **Qualitative results** produced by E-TSDM on Lsun-Cat $256 \times 256$ (Yu et al., 2015).

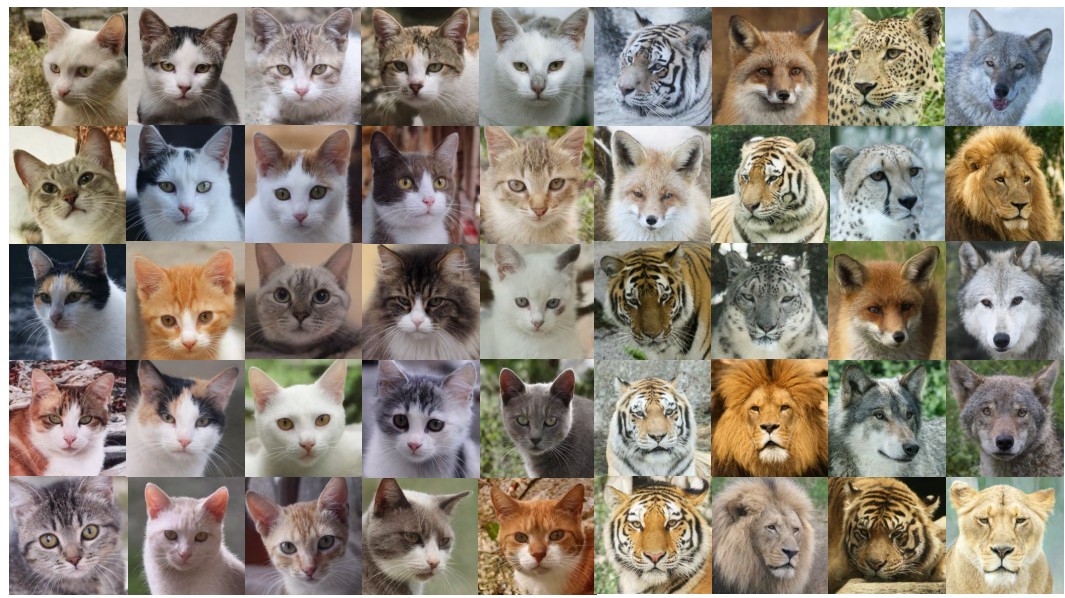

Figure A15: **Qualitative results** produced by E-TSDM on AFHQ-Cat $256 \times 256$ (Choi et al., 2020) and AFHQ-Wild $256 \times 256$ (Choi et al., 2020).

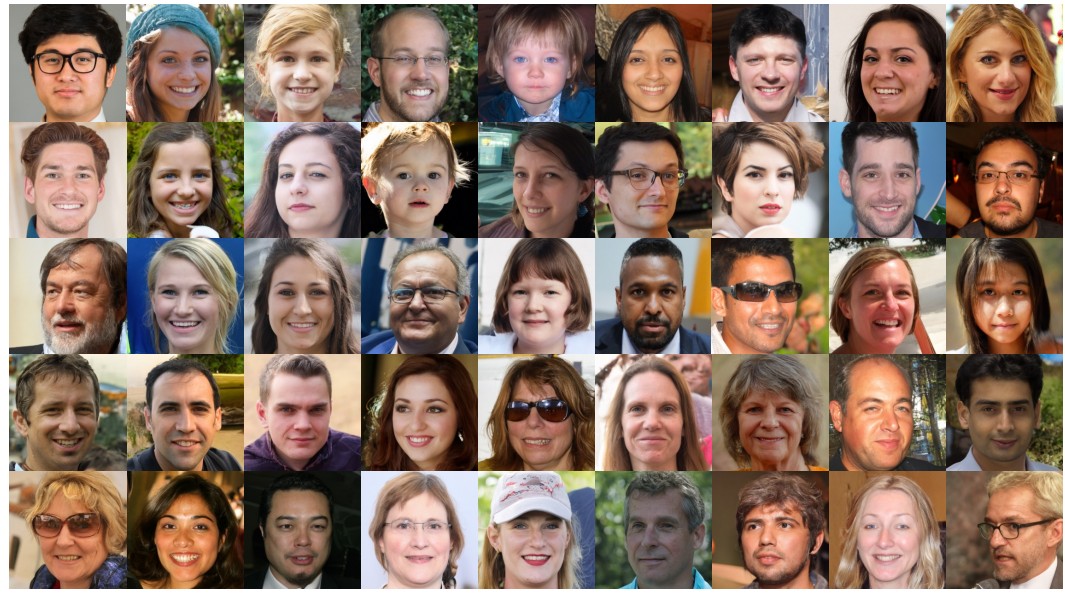

Figure A16: **Qualitative results** produced by E-TSDM on FFHQ $256 \times 256$ (Karras et al., 2019).

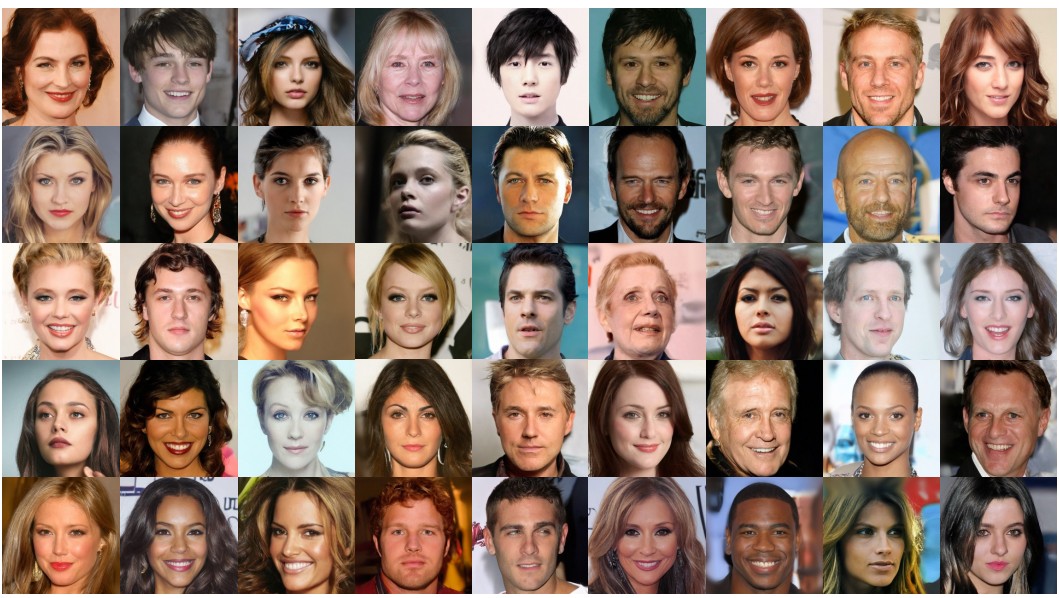

Figure A17: **Qualitative results** produced by E-TSDM on CelebAHQ 256 × 256 (Karras et al., 2017).

