# OpenReview forum: "Lipschitz Singularities in Diffusion Models"
_ICLR.cc/2024/Conference — ICLR 2024 oral_

### Official Review · Reviewer_gGRw · 2023-10-22

**Soundness:** 4 excellent
**Presentation:** 3 good
**Contribution:** 3 good
**Rating:** 6
**Confidence:** 4

**Summary:**

The paper elaborates upon an important observation concerning the presence of infinite-Lipschitz constants in the diffusion process, made earlier by (Song et al., 2021a; Vahdat et al., 2021). It also proposes a simple yet effective approach to address this challenge.

**Strengths:**

Theorem 3.1 is a nice piece of rigorous analysis of diffusion models, albeit indebted to Song et al.

The proposed approach to address this infinite-Lipschitz challenge, which is based on improving the resolution of the discretisation, does indeed seem to be effective.

Numerical results in Figures 3 and 4 seem quite impressive.

**Weaknesses:**

The observation concerning the presence of infinite-Lipschitz constants in the diffusion process is not original (Song et al., 2021a; Vahdat et al., 2021). Concerning it has been observed before, the authors should like to tone down their claims of having observed it first.

Some of the English is stilted ("vexing propensity of diffusion models" in the abstract, "Recently, there have been massive variants that significantly promote the development of diffusion models" on page 3).

**Questions:**

How would you describe the differences in your observation and those of (Song et al., 2021a; Vahdat et al., 2021)?

You could make your observation more original by noting that the infinite Lipchitz constants mean the SDE need not have a unique strong solution (Øksendal, 2003). Exhibiting multiple solutions would indeed be of interest.

**Details Of Ethics Concerns:**

None.

---

> ### Author Response · Authors · 2023-11-20
>
> Dear reviewer, we sincerely appreciate your valuable comments. Thank you for acknowledging the soundness of our research and the efficacy of our approach. Such recognition greatly boosts our confidence. Additionally, we are grateful for your insightful opinion, “the infinite Lipchitz constants mean the SDE need not have a unique strong solution”. We are deeply inspired by it.
>
> **Q1: The observation concerning the presence of infinite-Lipschitz constants in the diffusion process is not original. How would you describe the differences in your observation and those of (Song et al., 2021a; Vahdat et al., 2021).**
>
> **A1:** Thank you for reminding us of these related works. The aforementioned works indeed observed numerical instability issues near zero point, which are of great importance. **However, it is distinct from our work.** The numerical issues observed by previous works are mainly caused by the **singularity of the Gaussian transition kernel $q_{0t}(x_t|x_0)$ as $\sigma_t \rightarrow 0$**. However, our observation is *the infinite Lipschitz constants of the noise prediction model $\epsilon_\theta(x, t)$ w.r.t time variable $t$*, which are caused by **the explosion of ${\rm d}\sigma_t / {\rm d}t$, but not $\sigma_0=0$**. **To the best of our knowledge, the infinite Lipschitz constants are not pointed out by these works**. Besides, large Lipschitz constants near zero will pose a significant threat to both training and inference: during training, it can hurt the training on other timesteps due to the smooth nature of the neural network; during inference, it can affect the accuracy of numerical integration. **Such instability issues can not be well solved by the methods of previous works**. **Therefore, our work explores a different topic from previous works**. We believe the observation of the instability issues caused by the infinite Lipschitz constants, combined with our proposed solution to mitigate the instability, constitutes a good contribution to the diffusion model community.
>
> **Detailed analysis**
> While we have added further discussion in the related-work section (**Section 2**) of our paper, we also give a brief summary of the numerical issue observed by each related work and our work as below:
>
> **Song et al., 2021a:** In this work, it is mentioned in **section 4.3** that numerical instabilities exist when $t\rightarrow 0$. However, the authors don't point out what kind of numerical instability issue it is, and they don't provide any further analysis of the reasons behind the problem.
>
> **Vahdat et al., 2021:** In this work, it is pointed out in **Appendix G.4** that integration of probability flow ODE should be cut off close to zero since $\sigma_t$ goes to zero at $t=0$. However, there is no explanation why $\sigma_t \rightarrow 0$ makes integration cutoff necessary.
>
> **Song et al., 2021b:** This work is cited by Vahdat et al., 2021. To the best of our knowledge, this is the first work that proposes to cut off near zero during training and sampling for diffusion models. Specifically, it is pointed out in **Appendix C** that the variance of $x_t$ in the Gaussian perturbation kernel $p_{0t}(x_t | x_0)$ vanishes as $t \rightarrow 0$. This vanishing variance can cause numerical instability issues for training and sampling at $t=0$. Therefore, computation is restricted to $t \in [\epsilon, 1]$ for a small $\epsilon > 0$, *i.e.*, the cutoff strategy mentioned in Vahdat et al., 2021. Indeed, when the variance of $x_t$, denoted as $\sigma_t^2$, goes to zero, the Gaussian perturbation kernel $p_{0t}(x_t | x_0)=\mathcal{N}(x_t; \alpha_t x_0, \sigma_t^2 {\bf I}) \propto \frac{1}{\sigma_t^d}\exp(-\frac{1}{2\sigma_t^2}\Vert x(t) - \alpha_t x_0\Vert^2)$ will degrade to a Dirac kernel $\delta(x_t - \alpha_t x_0)$, and its log gradient $\nabla_{x_t} \log p_{0t}(x_t | x_0) = -\frac{1}{\sigma_t^2}(x_t - \alpha_t x_0)$ is not well defined when $\sigma_t=0$. As the score-based model $s_\theta(x, t)$ is directly optimized by denoising score matching with learning objective $\mathcal{L}_ t^{\rm score} = \mathbb{E}_ {x_ 0, x_ t}[\Vert s_ \theta(x, t) - \nabla_ {x_ t} \log p_ {0t}(x_ t | x_ 0)\Vert _ 2^2 ]$, an ill-defined $\nabla_{x_t} \log p_{0t}(x_t | x_0)$ at $t=0$ can cause numerical issues, as mentioned in Song et al., 2021b.

---

> ### Author Response · Authors · 2023-11-20
>
> **Our work:** Different from previous works, the focus of our work is the noise prediction model $\epsilon_\theta(x, t) = -\sigma_t \nabla_{x_t} \log p_t(x_t)$, and its learning objective at any $t$ is $\mathcal{L}_ t^{\rm noise}(\theta) = \mathbb{E}_ {x_0, \epsilon}[\Vert \epsilon_ \theta(\alpha_t x_0 + \sigma_t \epsilon, t) - \epsilon \Vert_ 2^2]$. Although $\mathcal{L}_ t^{\rm noise}$ only differs from $\mathcal{L}_ t^{\rm score}$ by a scalar transformation $\epsilon = -\sigma_t \nabla \log p_{0t}(\alpha_t x_0 + \sigma_t \epsilon | x_0)$ when $\sigma_t > 0$, $\mathcal{L}_t^{\rm noise}$ exhibits better numerical properties at $t=0$ since $\epsilon$ is just a sample of Gaussian distribution, and it is well defined when $\sigma_t=0$. From this point of view, noise prediction model should alleviate the numerical issues mentioned in Song et al., 2021b.  However, our analysis reveals that noise prediction model suffers from another numerical issue, *i.e.*, the infinite Lipschitz constants near zero. This problem is caused by the explosion of $\frac{{\rm d} \sigma_t}{{\rm d} t}$ near $t=0$, but not the vanishing variance $\sigma_t^2 \rightarrow 0$ as indicated in previous works.
>
> **Please let us know if there are any of your concerns, we are very willing to discuss them with you**
>
> **Q2: Some of the English is stilted.**
>
> **A2:** Thank you for incorporating the suggestions. Here's the revised version of the sentences:
> + "In this paper, we explore a perplexing tendency of diffusion models: they often display the infinite Lipschitz property of the network with respect to the time variable near the zero point."
> + "The significant advancements of diffusion models have been witnessed in recent years in the domain of image generation."
> Besides, we have relocated the second sentence to the beginning of the related works (Section 2) to enhance the logical flow and clarity.

---

> ### Author Response · Authors · 2023-11-20
>
> **Q3: You could make your observation more original by noting that the infinite Lipchitz constants mean the SDE need not have a unique strong solution (Øksendal, 2003). Exhibiting multiple solutions would indeed be of interest.**
>
> **A3:** Thank you for your kind suggestion. It is really an interesting problem. According to Theorem 5.2.1 (Existence and uniqueness theorem for stochastic differential equations) in the book you mentioned (Øksendal, 2003), for a general SDE ${\rm d} X_t = b(t, X_t){\rm d}t + \sigma(t, X_t){\rm d}B_t$, an unique strong solution exists if measurable functions $b(\cdot, \cdot)$ and $\sigma(\cdot, \cdot)$ are bounded and globally Lipschitz in state. Thus, if the infinite Lipschitz constants in time can break either of these two conditions, the SDE may not have a unique strong solution.
>
> However, in diffusion models, for any $\epsilon > 0$, a unique strong solution exists in $[\epsilon, T]$ since the above two conditions are satisfied. Multiple solutions can only exist at the zero point. Thus, we can cut off $[0, \epsilon]$ as proposed in previous milestone works (Song et al., 2021a,b; Vahdat et al., 2021), and this will not hurt the practical performance in most cases with a suitable $\epsilon$.
>
> It is worth noting that the cut-off $\epsilon$ should be small enough to avoid performance degradation. Thus, near $t=\epsilon$, the Lipschitz constants in time may still be large, and this can also lead to numerical instability in both training and inference. As demonstrated in our work, the instability issues can be significantly mitigated with our proposed method.
>
> All in all, the uniqueness of the solution is an important and interesting question, and thank you for your suggestion again. We will add this discussion to our paper.

---

> ### Author Response · Authors · 2023-11-20
>
> Dear reviewer, We would like to know if our response has dispelled all your concerns. If we have dispelled all your concerns, would you please raise our score? If there are any of your concerns, please let us know. We are very willing to discuss them with you.

---

> ### Author Response · Authors · 2023-11-22
>
> Dear reviewer, we notice that the response period is nearing its end. If you still have any concerns, we would greatly appreciate hearing from you. We eagerly await your response.

---

### Official Review · Reviewer_qCuR · 2023-10-30

**Soundness:** 4 excellent
**Presentation:** 3 good
**Contribution:** 3 good
**Rating:** 8
**Confidence:** 4

**Summary:**

This paper deals with an exploding Lipschitz constant in the function a neural network is asked to learn in a DDPM model, and the negative effects of trying to learn a function with such.

The authors present an argument based on taking time derivative of the quantity $-\sigma_t \nabla \log q_t(x)$, where the $\sigma_t$ are the standard deviations of the forward noising process in the time-discrete forward noising process in the DDPM formulation and $q_t$ is the density of the data distribution diffused to discrete time step $t$.

They demonstrate that the Lipshitz constant in time explodes to infinity for a most parameter settings of common noising schedules under the DDPM/VPSDE setting.

The authors propose a method for fixing this issue by applying a transform to the time input of the score network, tying together multiple timestep near t=0 to have the same score.

They demonstrate significant empirical benefit over a range of diffusion modelling tasks.

The authors also discuss a number of other possible methods to alleviate the issue of learning high lipshitz constants in diffusion models, but show that these methods despite being theoretically attractive, do not perform as well in practise.

**Strengths:**

1) The method proposed is simple to implement.
2) The method clearly demonstrates significant empirical benefit.
3) The authors discuss alternative proposals and show these are less effective

**Weaknesses:**

1) The only weakness I would like to highlight is the discussion of the alternative methods presented.
 - I believe 1 of the methods from the appendix is not mentioned in the main text - namely the Remp method (D.3.3).
 - It would be nice to see an expanded discussion of these with some small experiment to show the quantitative difference between the proposed method and these other methods. I appreciate the space limitation, but I think this is really an interesting point.

**Questions:**

1. Could the authors highlight better which lipshitz constant is is that is important, and why we care about it? While I understand I believe which and why it is cared about, it is perhaps not the clearest from reading the paper. The sentence in the abstract "they frequently exhibit the infinite Lipschitz near the zero point of timesteps" is a good example of this - it does not specify _what_ function has high lipshitz constant, or why indeed that matters. From reading the paper in depth, the authors care about the lisphitz constant of the quantity $-\sigma_t \nabla \log q_t(x)$ as a) neural networks find it difficult to learn high lipshitz constant functions, and this is the function we are asking the score net to learn, and b) because this quantity is involved in the reverse rollouts, having a term with high lipshitz constant makes discretising the SDE challenging to do accurately, but this should be apparent from the abstract/introduction.

---

> ### Author Response · Authors · 2023-11-19
>
> Dear reviewer, We extend our gratitude for acknowledging the significance of our research. Your commendation of our proposed method for its simplicity and efficacy, as well as your appreciation of our analysis of alternative methods, serve as a profound source of motivation for us. In light of your invaluable feedback, we have diligently incorporated several enhancements into our paper. We firmly believe that these refinements shall considerably augment the overall quality of our work.
>
> **Q1: "Remap" is not mentioned in the main text.**
> **A1:** Yes, thank you very much for reminding us. We didn't mention "Remap" in our main text because of space constraints. we have added a brief discussion of Remap in the "Comparison with some alternative methods" part of **Section 4**. This discussion aims to introduce the concept of Remap and highlight its limitations. Our intention is to draw the readers' attention to this intriguing alternative approach.
>
> **Q2: It would be nice to see an expanded discussion of these with some small experiment to show the quantitative difference between the proposed method and these other methods.**
> **A2:** Thank you for your advice, we have incorporated a new figure (**Figure 3** of the updated PDF) in the "Comparison with some alternative methods" part of **Section 4**, to visually represent the quantitative evaluation of these alternative methods. By including this visual representation, we aim to facilitate a rapid understanding of the performance of these alternative methods for the readers.
>
> **Q3: Could the authors highlight better which lipshitz constant is important, and why we care about it?**
> **A3:** Thank you for your valuable suggestion. We have made the necessary adjustments in the **abstract** and **introduction** by explicitly mentioning which specific Lipschitz constants we are concerned with. Additionally, we have included a concise discussion in the **introduction** to elucidate the importance of these Lipschitz constants. Our aim with these modifications is to enhance the readers' comprehension of our work and minimize any potential confusion.

---

> ### Author Response · Authors · 2023-11-20
>
> Dear reviewer, We would like to know if our response has dispelled all your concerns. If there are any of your concerns, please let us know. We are very willing to discuss them with you.

---

### Official Review · Reviewer_DUEU · 2023-10-31

**Soundness:** 4 excellent
**Presentation:** 4 excellent
**Contribution:** 3 good
**Rating:** 8
**Confidence:** 3

**Summary:**

This paper demonstrates theoretically (and confirms empirically) that the limit of the Lipschitz constant of the noise prediction network for a timestep of zero is infinite. Such a result is a source of instability for using diffusion models in many generative tasks, and the authors propose a technical solution to alleviate this issue and confirm the superiority of the approach with extensive numerical simulations.

**Strengths:**

This is an excellent paper, and the presentation is very well carried out. The authors point out a very interesting theoretical property that could explain some practical instabilities encountered in DDPM samples. They then present a practical solution to the problem. The authors' contribution is excellent for the community, as reducing the instabilities in the generative process, such as diffusion models, has important practical consequences.

**Weaknesses:**

This paper as it is impeccable in terms of presentation and contribution, both theoretically and practically. The only drawback is that no open-source code is available to experiment with their approach.

**Questions:**

None

---

> ### Author Response · Authors · 2023-11-19
>
> We greatly appreciate your kind words and recognition of our work. We are highly encouraged by your kind words "The authors' contribution is excellent for the community, as reducing the instabilities in the generative process, such as diffusion models, has important practical consequences." We have anonymously submitted the core part of our code as supplementary material and will make it publicly available once the paper is accepted. Thank you once again for your support.

---

### Official Review · Reviewer_gj5E · 2023-11-01

**Soundness:** 4 excellent
**Presentation:** 3 good
**Contribution:** 4 excellent
**Rating:** 8
**Confidence:** 3

**Summary:**

Diffusion models, utilizing stochastic differential equations to generate images, have become a leading type of generative model. However, their underlying diffusion process hasn't been thoroughly examined. This paper reveals a concerning tendency in diffusion models: they often display infinite Lipschitz (for $\sigma_{t} \cdot \text{score function}$) near the initial timesteps. Through theoretical and empirical evidence, the presence of these infinite Lipschitz constants is confirmed, which can jeopardize the stability and precision of the models during training and inference. To combat this, the paper introduces a new method, E-TSDM, that uses quantization to reduce these Lipschitz issues. Tests on various datasets support the presented theory and approach, potentially offering a deeper understanding of diffusion processes and guiding future diffusion model design.

**Strengths:**

This paper highlights a unique and previously unexplored challenge with DDPM: the instability encountered when learning $\epsilon_{\theta} = \sigma_{t} \cdot \nabla \log q_{t}(x)$ during the time steps where $\sigma_{t}$ is minimal. One might naturally question why DDPM doesn't directly learn $\nabla \log q_{t}(x)$. I conjecture that the optimization process for learning $\nabla \log q_{t}(x)$, which involves solving $E\|\nabla \log q_{t}(x) - \frac{1}{\sigma_{t}} \|^2$, becomes problematic with a small $\sigma_{t}$. As a workaround, DDPM employs a transformation to learn $\sigma_{t}\cdot \nabla \log q_{t}(x)$ directly. However, this paper reveals the inherent price of such an approach (no free lunch indeed).

The paper validates the infinite Lipschitz problem with $\epsilon_{\theta}$ both theoretically and empirically. Moreover, it introduces E-TSDM, an innovative solution that essentially employs a quantization strategy when $\sigma_{t}$ is minimal, particularly during the initial t=100 steps. Comprehensive experiments demonstrate E-TSDM's enhanced stability and performance, even setting a new benchmark for FFHQ 256×256.

The paper's novelty is commendable, presenting a compelling and succinct argument with an impressive practical performance. Its insights could significantly influence the diffusion model community. I'm inclined to strongly endorse its acceptance.

**Weaknesses:**

- One minor suggestion is to avoid saying $t$ being small (rather, it is about $\sigma_{t}$ being small). Since $t$ is in fact $0, 1, 2, 3, .. 100.$
- May add more discussions to the alternative approaches (see Questions below).
- It may be worth showing that directly learning $\nabla \log q_{x}(t)$ with the least square is prohibitve.

**Questions:**

I am looking for comments from the authors on a few alternative methods:
1. Learning $\nabla \log q_{x}(t)$ directly with weighted least square: can we reduce the weight of the least square when $\sigma_t$ is small, e.g., learn $E \sigma_{t}^2\|\nabla \log q_{x}(t) - \frac{1}{\sigma_{t}}I\|^2$?
2. For Eq (9), what if we only learn $\epsilon_{\theta}(\alpha_{f_T(t)}x_0 + \sigma_{f_T(t)}\epsilon, f_{T}(t))$, i.e., only learn the score function for time $f_{T}(t)$ and only use those time steps to do sampling?
3. Is that $\sigma_{t}$ an input of the neural network: what if we learn $\epsilon_{\theta}(x, \sigma_{t})$ (the intuition is that $\sigma_{t}$ will help adjust the network Lipschize automatically).

Minor comments:
- Eq 10, are $\beta_{t}$ and $\eta_{t}$ defined?

---

> ### Author Response · Authors · 2023-11-19
>
> Esteemed reviewer, we express our heartfelt gratitude for your recognition of the novelty, effectiveness, and contribution of our work. Additionally, we extend our utmost appreciation for your invaluable advice. We firmly believe that these suggestions hold the potential to enhance the quality of our paper.
>
> **1. This work reveals the inherent price of predicting noise.**
> Your astute observation regarding our work's revelation of the inherent cost associated with predicting noise instead of directly learning $\nabla \log q_ t(x)$ is greatly appreciated. We wholeheartedly concur with your insightful perspective. In response to this, we have included detailed discussions about this viewpoint within **Section 2** (Related Works) of our paper. Specifically, we begin by introducing the challenges associated with directly learning the score function, followed by an exposition on commonly employed noise-prediction models. Ultimately, we emphasize that our research reveals the inherent drawbacks of noise-prediction models.
>
> **2. Reply for questions about alternative methods**
> Following your recommendations, we have incorporated four additional experiments on FFHQ $256\times 256$ into our research. The evaluation metric used for these experiments is FID-10k. To ensure fairness, we have maintained the same experimental settings as those employed in the baseline.
>
> **Q1: Directly learning $\nabla \log q_t(x)$ with least square and with weighted least square.**
> **A1:** In accordance with your analysis, directly learning $\nabla \log q_t(x)$ with least square presents challenges, especially with a small $\sigma_t$. We have implemented this approach and present a quantitative comparison in the table below. The experimental results affirm that directly learning $\nabla \log q_t(x)$ using the least square method is not feasible.
>
> Moreover, we have also attempted to learn $\nabla \log q_t(x)$ using a weighted least square approach. Specifically, we aimed to learn $\theta^* = \arg\min_ {\theta} \mathbb{E}_ t \left[ \sigma_t^2\mathbb{E}_ {x}\left[ \Vert s_ \theta(x, t) - \nabla \log q_ {0t}(x|x_0) \Vert_2^2\right] \right] $. Intuitively, the weighted least square method should outperform the original least square method, as it mitigates the influence of problematic intervals. As depicted in the subsequent table, learning $\nabla \log q_t(x)$ with the weighted least square method yields significantly superior results compared to direct learning with the least square method.
>
> **Table1. Quantitative comparison between directly learning $\nabla \log q_t(x)$ with the least square and the baseline on FFHQ $256\times 256$.**
> | Method | FID-10k |
> | --- | --- |
> | Learning $\nabla \log q_t(x)$ with LS | 97.96|
> | Learning $\nabla \log q_t(x)$ with weighted LS | 13.87 |
> | Baseline | 9.50 |
>
> **Q2: What if we only learn the score function for time $f_T(t)$ and only use those time steps to do sampling.**
> **A2:** This method exhibits inferior performance compared to the baseline. The experimental results for FFHQ $256\times 256$ are presented in the table below. Specifically, when $t \ge 100$, we train the score function for $t=100, 101, \dots, T-1$, and when $t < 100$, we only train the score function for $t = 0, 20, 40, 60, 80$. During inference, we exclusively employ the timesteps of $f_T(t)$ for sampling. Likewise, the baseline method also restricts sampling to the timesteps of $f_T(t)$. The experimental results, depicted in the subsequent table, demonstrate the inferior performance of this method compared to the baseline.
>
> Although sampling in our method is limited to the timesteps of $f_T(t)$ ($t = 0, 20, 40, 60, 80$ when $t < 100$), training on additional timesteps can provide valuable information and enhance the network's denoising capabilities. By solely learning from the timesteps of $f_T(t)$, the network struggles to capture information from other timesteps, resulting in subpar performance. On the other hand, if we adopt our proposed method to train on all timesteps but limit sampling to the timesteps of $f_T(t)$ , then it becomes the situation of DDIM sampling. The outcomes of DDIM are presented in **Table 3** of our paper. However, if we extend the sampling to all timesteps, it will further enhance the performance.
>
> **Table2. Quantitative comparison between only training for time $f_T(t)$ and the baseline on FFHQ $256\times 256$.**
> | Method | FID-10k |
> | --- | --- |
> | Training on all $f_T(t)$, sampling on $f_T(t)$ | 48.15|
> | Training on all $t$, sampling on $f_T(t)$ (Baseline) | 21.75 |

---

> ### Author Response · Authors · 2023-11-19
>
> **Q3: What if we learn $\epsilon_\theta(x, \sigma_t)$.**
> **A3:** Based on our understanding, this method aims to replace the network's conditional input $t$ with its corresponding $\sigma_t$. If there are any misunderstandings, please let us know.
>
> **1) This method can be classified as "Remap" mentioned in our paper.**
> In reality, this method can be classified as one of the alternative methods mentioned in our paper, *i.e.*, Remap (Please refer to the "Comparison with some alternative methods" part in **Section 4** and **Section D.3.3** in the appendix). The core idea behind Remap is to design a remap function $\lambda=f(t)$ as the network's conditional input instead of directly utilizing $t$, *i.e.*, $\epsilon_\theta(x_t, \lambda)$. In this particular case, we set $\lambda = \sigma_t$.
>
> **2) Analysis of Remap, where the remap function is $\lambda = \sigma_t$**
> The performance of Remap relies heavily on whether sampling is performed uniformly based on $t$ or $\lambda$ during both training and inference. **1) Uniformly sampling $t$:** As discussed in our paper, if the sampling strategy remains consistent (uniformly sampling $t$) during both training and inference, Remap has no impact on $\frac{\partial \epsilon_\theta(x_t, t)}{\partial t}$. As a result, Remap may yield similar performance to the baseline. **2) Uniformly sampling $\lambda:$** However, if there is a change in the sampling strategy (uniformly sampling $\lambda=f(t)$, i.e. $\sigma_t$ here) during training **or** inference, the inference results may deteriorate, similarly to the examples shown in our paper. This occurs because the equivalent schedule may compel the network to focus on specific stages of the entire process. For instance, taking $\lambda=\sigma_t$ as an example, where $\sigma_{999} = 1.0000$, $\sigma_{782}= 0.9990$, and $\sigma_{0}= 0.01$. If we uniformly sample $\sigma_t$ during training, timesteps falling within the range $t\in(782, 999)$ will rarely undergo training. We provide additional examples of remap functions, with a quantitative evaluation provided in **Figure 3**, along with a detailed analysis in the "Comparison with some alternative methods" section in **Section 4** and **Section D.3.3** in the appendix.
>
> Besides, it is important to note that $\lambda=\sigma_t$ is a special remap function. The difference of $\sigma_{t}$ and $\sigma_{t-1}$ gets extremely small near $t=T$. This characteristic may hinder the network's training process, as it is hard for the network to distinguish adjacent timesteps with extremely similar conditional inputs. Consequently, learning $\epsilon_\theta(x, \sigma_t)$ may yield poorer performance than the baseline, as confirmed by the experimental results presented in the following table.
>
> **Table3. Quantitative comparison between using $\sigma_t$ as conditions and the baseline on FFHQ $256\times 256$.**
> | Method | FID-10k |
> | --- | --- |
> | $\epsilon_\theta(x, \sigma_t)$ | 16.41|
> | Baseline | 9.50 |
>
>
> **3. Reply for other questions**
>
> **Q4:Avoid saying t being small.**
> **A4:** Thank you, we appreciate your reminding. To prevent any potential misunderstandings, we replace "small $t$" with "small $\sigma_t$".
>
> **Q5: Eq10, are $\beta_t$ and $\eta_t$ defined?**
> **A5:** Yes, thank you for pointing out this oversight. we have included their definition ($\beta_t = 1 - \frac{\alpha_t}{\alpha_{t-1}}$, and $\eta_t^2 = \beta_t$.) after Equation (10).

---

> ### Author Response · Authors · 2023-11-20
>
> Dear reviewer, We would like to know if our response has dispelled all your concerns. If there are any of your concerns, please let us know. We are very willing to discuss them with you.

---

> > ### Comment · Reviewer_gj5E · 2023-11-23
> > **Thank you!**
> >
> > I am grateful for the authors' comprehensive and insightful responses. They took great care in addressing my questions and the results highlighted their contribution again.

---

### Author Response · Authors · 2023-11-19

We extend our sincere appreciation to all the reviewers for their invaluable feedback, which greatly aids in refining our work. The sound theory and effective proposed method have received high praise from all reviewers. As you mentioned, we believe that our work makes a good contribution to the community by addressing the instability commonly associated with diffusion models. Your support is of utmost importance to us and deeply cherished.

**Summary of revision**

In response to the valuable recommendations we received, we have made the following modifications to our paper:
+ We have **incorporated** a new discussion on the "remap" method (which is previously discussed in the appendix) into the "Comparison with some alternative methods" part of **Section 4**.
+ We have **included** a new figure (**Figure 3**) showcasing quantitative evaluations of the three mentioned alternative methods (which is previously reported in the appendix), to enhance reader comprehension of their performance.
+ We have **expanded** the discussion in the "numerical stability near zero point" part of **Section 2** (Related work). Specifically, we have provided a more detailed description of the disparities between our findings and those of prior works. This extension aims to provide readers with a better understanding of the contribution of our research.

To save space:
+ We have **relocated** **Figure 2** (The comparison of Lipschitz constants in continuous-time scenarios) from the original paper to the appendix (**Figure A1**). This decision was made as Figure 2 closely resembles Figure 1(b) except for the settings.
+ We have **moved** **Figure 6** (The result of the super-resolution task) from the main paper to the appendix (**Figure A12**). However, to maintain clarity within the super-resolution task section, we have retained the quantitative results in the main paper and left the previous analysis unchanged.
+ We have made additional minor modifications to ensure compliance with the 9-page limit.

These revisions, guided by the reviewers' invaluable insights, serve to enhance the overall quality and readability of our paper. We highlight all revisions in blue.

---

### Public Comment · ~Haowei_Zheng1 · 2023-12-13
**Two Questions About Proofs in This Paper**

I have two questions about this paper.

1. In Theorem 4.1, this paper assumes that $B(x):=\sup\limits_{t}\Vert\nabla_{x}\log q_t(x)\Vert < \infty$. However, many previous works [1] [2] [3] have shown that the score function $\nabla_x \log q_t(x)$ asymptotically tends to infinity as $t \rightarrow$ 0, i.e., $\lim\sup\limits_{t \rightarrow 0+}\Vert\nabla_x \log q_t(x)\Vert=\infty$.
2. In theorem 3.1, this paper assumes that $q_t(x)$ is a smooth process such that $\lim\sup\limits_{t\to0+}\Vert\frac{\partial\nabla_\mathbf{x}\log q_t(\mathbf{x})}{\partial t}\sigma_t\Vert<\infty $. But I am not sure how to make such a conclusion given this assumption.

Could the authors provide further clarification or justification for these assumptions?

[1] NeurIPS2022-Score-Based Generative Models Detect Manifolds

[2] ICML2023-Score Approximation, Estimation and Distribution Recovery of Diffusion Models on Low-Dimensional Data

[3] TMLR2022-Convergence of denoising diffusion models under the manifold hypothesis

---

### Meta-Review · Area_Chair_JqEU · 2023-12-11

**Metareview:**

**Summary**

This paper identify the issue of infinite Lipschitz constant in the standard DDPM denoising networks both theoretically and empirically, which leads to instability for both training and inference.
Then, a new method is proposed to reduce these Lipschitz issues which are widely range of diffusion modeling tasks with significant empirical benefits.


**Strengths**

1. The paper theoretically shows the Lipschitz singularity issue in standard noise prediction networks, which was previously unexplored.

2. The proposed E-TSDM shows significant improvement in migrating the instability issue.


**Weaknesses**

1. Missing additional ablations on alternative learning objectives as suggested by the reviewers.

2. Missing clarification on the importance of Lipschitz constant

3. Writing can be improved to avoid confusion

**Justification For Why Not Higher Score:**

N/A

**Justification For Why Not Lower Score:**

The paper received uniformly positive feedback from all the reviewers. The sound theory and effective method significantly contribute to addressing the instability issue of diffusion models.

---

### Decision · Program_Chairs · 2024-01-16

Accept (oral)